



# Aerosol optical properties within the atmospheric boundary layer predicted from ground-based observations compared to Raman lidar retrievals during RITA-2021

Xinya Liu[1], Diego Alves Gouveia[2], Bas Henzing[3], Arnoud Apituley[2], Arjan Hensen[3], Danielle van Dinther[3], Rujin Huang[4], Ulrike Dusek[1]

[1]Centre for Isotope Research (CIO), Energy and Sustainability Research Institute Groningen (ESRIG), University of Groningen, Groningen, 9747 AG, the Netherlands
[2]Royal Netherlands Meteorological Institute (KNMI), De Bilt, 3730 AE, the Netherlands
[3]Department of Climate, Air and Sustainability, TNO, Utrecht, 3584 CB, the Netherlands
[4]State Key Laboratory of Loess and Quaternary Geology, Center for Excellence in Quaternary Science and Global Change, and Key Laboratory of Aerosol Chemistry and Physics, Institute of Earth and Environment, Chinese Academy of Sciences, Xi'an 710061, China

*Correspondence to*: Ulrike Dusek (u.dusek@rug.nl)

## Abstract

In this study, a Mie theory-based model was built to predict the vertical profile of the aerosol optical properties, including the aerosol scattering coefficient, backscatter coefficient, extinction coefficient, and lidar ratio. The model utilizes ground-based in-situ measurements of the aerosol chemical composition and particle size distribution, as well as the meteorological data from the Weather Forecasts (ECMWF) as input values. These are all parameters readily obtained for ACTRIS sites and the aim of this study was to investigate their suitability for generating representative estimates of the lidar ratio, and then further improve the lidar retrievals by utilizing these estimates. The measurements were performed during the Ruisdael land-atmosphere interactions Intensive Trace-gas and Aerosol (RITA) campaign in the Netherlands in 2021. The calculated dry aerosol optical properties were validated against a Nephelometer with good agreements ($R^2 \approx 0.9$). The predicted ambient vertical profiles of aerosol optical properties were compared to retrievals by a multi-wavelength Raman lidar. Predicted and retrieved backscatter coefficients were usually comparable under conditions of a well-mixed boundary layer. The extinction coefficients and lidar ratios were retrieved by the Raman lidar only at a height above 800 m. The estimated lidar ratio profiles based on in-situ data connected reasonably well to the lidar profiles within the boundary layer, with differences on average $\pm$ 30%. Our study shows that for well-mixed boundary layers, a representative lidar ratio can be estimated based on ground-based in-situ measurements of dry size distribution and chemical composition taking into account the hygroscopic growth and ambient humidity. This allows to extend extinction profiles to lower altitudes, where they cannot be retrieved, or for use with simple elastic backscatter lidar to derive extinction profiles. The proposed method could be further applied to predict aerosol optical depth and also might be beneficial for large-scale or global radiation simulations.



# 1    Introduction

Aerosols play an important role in climate change by altering the earth's radiation budget through their interaction with solar radiation. Aerosols reflect part of the sunlight, thereby reducing the radiation at the earth's surface (Twomey, 1977; IPCC, 2013), which results in a cooling effect. On the other hand, certain types of aerosols can also absorb solar radiation, which locally warms the atmosphere and results in a change of the temperature profile, further affecting the atmospheric circulations (Koren et al., 2008; Rosenfeld et al., 2014; Bréon, 2006). In addition, aerosol particles can act as cloud condensation nuclei or ice nuclei affecting the microphysical properties of clouds, and thereby affect the radiation budget indirectly (Graf, 2004; Lohmann and Feichter, 2005; Bréon, 2006). There are still large uncertainties in predicting the contribution of aerosol radiative forcing to climate change, due to the complexity of microphysical and chemical process and their dynamic feedback on the aerosol budget (Kaufman et al., 2005; Feingold, 2001; Graf, 2004). To reduce the uncertainties, observation and simulation of aerosol optical properties and their vertical profiles are essential for a better understanding of aerosol radiative forcing (Moise et al., 2015; Chang et al., 2006).

Light detection and ranging (lidar) is a widely used active remote sensing method for studying the spatial distribution aerosol optical properties (Sicard et al., 2011; Kim et al., 2007; Bhardwaj et al., 2016; Markowicz et al., 2016). The detected signal of the elastically backscattered light can be converted into the backscatter and extinction coefficients based on an analytical solution of the so-called "lidar equation" (Klett, 1981; Fernald, 1984) with the assumption of a given extinction to backscatter ratio, called the "lidar ratio". However, the lidar ratio is governed by many factors such as the wavelength of incoming light, the aerosol chemical composition, particle size distribution, relative humidity, and other atmospheric conditions (Noh et al., 2008; Balis et al., 2004; Lopes et al., 2013; Shin et al., 2018; Dawson et al., 2015; Moise et al., 2015). Large errors can occur when retrieving aerosol extinction from backscattered signals. Thus, a Raman lidar technique based on Raman spectroscopy was developed to address this problem (Ansmann et al., 1990). The profiles of the backscatter and the extinction coefficient can be determined independently by the Raman lidar, without the assumption of a lidar ratio (Cooney et al., 1969; Melfi, 1972; Ansmann et al., 1990, 1992). However, a common limitation on the accuracy of the lidar-based retrievals emerges for distances close to the instrument where only a fraction of the atmospheric volume illuminated by the laser pulse is within the lidar's receiver field-of-view, resulting in a "blind zone" at the instrument (no overlap) and a region that is gradually becoming visible for the receiver after some distance (incomplete overlap region) (Wandinger, 2005). While Raman backscatter retrials can be less affected by the incomplete overlap region, Raman extinction and elastic lidar retrievals are particularly sensitive to it even after an overlap correction is applied, and thus it can only accurately record the aerosol profiles above a certain altitude (Rosati et al., 2016; Hervo et al., 2016; Wandinger and Ansmann, 2002).

Besides active remote sensing, vertical aerosol profiles can also be measured by in situ airborne instruments (Düsing et al., 2021; Haarig et al., 2019; Düsing et al., 2018). These give more accurate information, but are expensive and time consuming and thus lack the temporal coverage of lidar measurements. They are essential in the evaluation of the lidar retrievals and several studies have modelled the aerosol optical vertical profiles based on the Mie theory using the vertically resolved aerosol



information but measured by the airborne instruments (Düsing et al., 2021; Ferrero et al., 2019; Düsing et al., 2018). Their results support the usefulness of in situ observations for evaluation of lidar retrievals, however, there are only a few profiles available due to the high cost of the airborne measurements.

In this study, we evaluate a method to predict vertical profiles of the aerosol optical properties using ground-based in situ measurements of the aerosol chemical composition and particle size distribution combined with meteorological profiles from
70 the European Centre for Medium-Range Weather Forecasts (ECMWF). The experiments were performed at Cabauw Experimental Site for Atmospheric Research (CESAR) site in the Netherlands during the Ruisdael land-atmosphere interactions Intensive Trace-gas and Aerosol (RITA) 2021 field campaign[1]. The primary goal of this study is to evaluate if routine ground-based measurements can be used to predict the lidar ratio and the extinction coefficient in the lower atmosphere, where it cannot be retrieved by the Raman lidar. If successful, this information can then be used to extrapolate extinction
profiles to the ground or to derive extinction data from elastic backscatter lidars. A further goal is to explore under which circumstances the aerosol measured on the ground can represent the vertical aerosol distribution in the atmosphere. The advantage of the proposed method is that we use only ground-based data that are readily available at most lidar sites and the easily obtained ECMWF data.

## 2 Methods

### 2.1 Experiment site and campaign description

The RITA campaign was carried out at the CESAR in the Netherlands (51.97° N, 4.93° E) in 2021. CESAR is one of the core observatories for the Ruisdael[2] observatory and also part of the ACTRIS[3] (Aerosol, Clouds and Trace Gases Infra-Structure) and ICOS[4] (Integrated Carbon Observation System). An aerial view of the infrastructure setup during the RITA campaign and the CESAR location are shown in Figure S1. The site is situated in a polder 0.7 m below average sea level and surrounded by
85 a flat pasture landscape. The mode of the wind direction distributions was southwest, but winds were also coming from the northeast as shown in Figure S2, so the potential pollution sources could be from Rotterdam with its large international harbour but also from nearby Utrecht. The ground-based aerosol in situ measurements included the aerosol chemical composition, particle size distribution and aerosol optical properties. Two intensive measurement campaigns were performed during Spring (11th May - 24th May) and during Fall (16th September - 12th October). The remote sensing observations by the Raman lidar
were obtained regularly during the campaign depending on the atmospheric conditions.

---

[1] https://ruisdael-observatory.nl/the-rita-2021-campaign/
[2] https://ruisdael-observatory.nl/ (last access: 20 July 2022)
[3] http://actris.net/ (last access: 20 July 2022)
[4] https://www.icos-cp.eu/ (last access: 20 July 2022)



### 2.1.1 Aerosol chemical composition measurements

Aerosol chemical composition was measured by different online and offline methods during the campaign.

(i) A time of flight-aerosol chemical species monitor (TOF-ACSM; Aerodyne Research Inc., Billerica, MA) equipped with capture vaporizer (CV) and $PM_{2.5}$ lens measured the mass concentration of non-refractory chemical compounds with a 10-minute time resolution. The TOF-ACSM was installed in a trailer, which was next to the remote sensing site as shown in the Figure S1(a) approximately 200 m distant from the other in situ measurements. The inlet was a Teflon Coated Aluminum cyclone (URG 2000-30ED) with an aerodynamic cut-off diameter of 2.5 µm at ambient conditions and the inlet flow rate was 2.3 L min$^{-1}$ controlled by the ARI Sample Line Flow Controller (S/N FCB-023) at the head of the TOF-ACSM inlet. Particles were dried by a Nafion dryer (Perma Pure, New Jersey). Five chemical species, namely ammonium ($NH_4^+$), nitrate ($NO_3^-$), sulphate ($SO_4^{2-}$), chloride ($Cl^-$), and organics (Org), were derived based on the fragmentation tables for TOF-ACSM (Fröhlich et al., 2013). The standard calibrations, such as the flow rate calibration, lens calibration and heater bias (HB) voltage tuning were performed before and after the campaign. Ionization efficiency (IE) and the relative ionization efficiency (RIE) were determined by calibration with $NH_4NO_3$ and $(NH_4)_2SO_4$ solutions with a concentration of 0.005 Mol L$^{-1}$. The calibration values used in this study are: IE $NO_3$ = 258.20 pg s$^{-1}$; RIE $NH_4$ = 3.51; RIE $SO_4$ = 1.33; RIE Org =1.40; RIE Chl=1.30; at an air beam (AB) = 4.55E + 5 ions s$^{-1}$; flow rate = 1.46 cm$^3$ s$^{-1}$. The data were processed by the Tofware software (version of 3.2.4, Tofwerk AG, Thun, Switzerland).

(ii) $PM_{2.5}$ and $PM_{10}$ filter samples were collected for 24 hours using a SEQ47/50 (Leckel GmbH, Germany) instrument with a sequential low-volume system (LVS) of 2.3 m$^3$ h$^{-1}$ next to the trailer with the TOF-ACSM. The sampler operation was based on the European Standards (EN12341: 1998 and EN14907: 2005). The filter samples were collected under ambient conditions, stored at approximately -20°C, and protected using ice packs during transportation. The concentrations of 3 inorganic anions ($NO_3^-$, $Cl^-$, $SO_4^{2-}$) and 5 cations ($Na^+$, $K^+$, $Mg_2^+$, $Ca_2^+$, $NH_4^+$) were determined by chromatography (ICS-1100, Thermo Scientific). Organic carbon (OC) and elemental carbon (EC) were analysed by a Sunset thermal optical analyzer (TOA, Sunset Laboratory Inc.) using the EUSAAR2 protocol (F. Cavalli, M. Viana, K. E. Yttri, J. Genberg, 2010; Karanasiou et al., 2020). The details of the data evaluation can be found in Liu et al., (2023).

(iii) the equivalent Black Carbon (eBC) mass concentration was measured by the multi-angle absorption photometer (MAAP model 5012, Thermo Fisher Scientific Inc., Franklin, MA) with 5 minute time resolution (Petzold and Schönlinner, 2004; Petzold et al., 2005). A constant scattering cross section value (6.6 m$^2$ g$^{-1}$) based on the user handbook was given for converting the aerosol light absorption coefficient at 670 nm.

The MAAP and the other in situ measurements discussed below were installed in the Cabauw main building, underneath the 213 m high tower as displayed in Figure S1. The MAAP was measuring behind a $PM_{10}$ inlet that was situated 4.5 m above the ground on the roof. A wide diameter Nafion drying system were installed after the $PM_{10}$ size selector to dry the ambient aerosol to an RH below 40%. After the Nafion a manifold split the aerosol flow equally to the multiple instruments. An overview of



the chemical composition and meteorological information of the in-situ measurements from the May to November in 2021 is displayed in the Figure S3.

### 2.1.2    Particle size distribution measurements

The particle number size distribution (PNSD) was measured by a Scanning Mobility Particle Size Spectrometer (MPSS, TROPOS) and an Aerodynamic particle sizer spectrometer (APS, Model 3321, TSI) which were connected to the same inlet as the MAAP. The MPSS measures particles in the size range from ~10 to 800 nm in electromobility diameter with a time resolution of 5 minutes. Before entering the MPSS, the particles are dried to below 40% relative humidity (RH) by a Perma Pure Nafion air dryer and then charged by a bipolar particle charger (Ni-63). The recorded data was inverted by a custom evaluation software (DMPS-Inversion-2.13.exe) correcting for the diffusion losses of the particles, bipolar charge equilibrium, and the DMA transfer function, as well as the CPC counting efficiency (Wiedensohler et al., 2012). The APS (Peters and Leith, 2003) covers an aerodynamic size range from 0.5 to 20 µm with data recorded in 1-minute time resolution. However, due to the inlet size cut off, the valid size range of the APS is from 0.5 to 10 µm.

The MPSS and APS measured size distributions were merged to create a particle size distribution with a diameter range from 10 nm to 10 µm following the method of Modini et al.(2021). We used the hourly merged particle size distribution to calculate the optical properties for a 5-month period and then compared with the nephelometer measurement. To calculate the vertical profiles, the PNSD data is averaged at a time resolution of 10 minutes. Subsequently, the nearest time period within the radar measurement range is selected for averaging. In this study, the MPSS electrical mobility diameters were assumed to correspond to volume-equivalent diameter, then APS aerodynamic diameters were converted to volume equivalent diameters. However, shape effects were neglected. The details of joining the PNSD are described in the supplementary material in section S2 as shown in Figure S4.

### 2.1.3    Ground-based measurements of aerosol optical properties

The aerosol scattering and backscatter coefficient were measured with a 5-minute time resolution by a three-wavelength integrating nephelometer (Dry Neph, TSI Inc., Model 3563) located in the main building adjacent to the MAAP. The Nephelometer measured the aerosol scattering coefficient using a wide angular integration (from 7 to 170°) and the backscatter integrated from 90 to 170° (Anderson et al., 1996; Anderson and Ogren, 1998; Heintzenberg and Charlson, 1996). Scattering coefficients integrated from 0 to 180° were derived based on the truncation correction function proposed in Anderson and Ogren (1998). The truncation error ranges from approximately 5% to 10% for submicron particles and from 30% to 50% for particles between 1 and 10µm (Anderson and Ogren, 1998; Anderson et al., 1996; Muller et al., 2009).



## 2.2 Meteorological observations

The meteorological data used in this study are obtained from the ACTRIS data portal[5], which are the Near Real Time (NRT) data generated by the ECMWF IFS forecast Model with 1-hour time resolution. Figure S5 shows the RH and temperature profiles derived from the ECMWF model from May to November in 2021. In situ measured meteorological parameters at different heights (7 m, 10 m, 20 m, 40 m, 80 m, 140 m, 200 m) were also recorded cat the 213 m high mast of CESAR tower with a 10-minute time resolution. Data available from May to June in 2021 and can be requested from the KNMI Data Platform[6]. However, we need meteorological profiles that cover the boundary layer depth reaching far beyond the tower height. A radiometer (RPG-HATPRO) located at the CESAR remote sensing site provided vertical profiles of RH and temperature from May to October in 2021. In addition, in-situ measurements of meteorological data were provided by a radiosonde (Vaisala RS92-SGP) carried on a balloon, which was launched every day at around 00:00 UTC from the De Bilt, approximately 25 km from the CESAR site. Previous studies (Fernández et al., 2015; Apituley et al., 2009) concluded that the atmospheric conditions at the CESAR observatory and at the De Bilt site are not significantly different. Therefore, the in-situ measurements from radiosonde were used to evaluate the meteorological profiles during the campaign period (in Figure S6). The findings demonstrated that the ECMWF data closely align with the in-situ measurements from the radiosonde by the balloon. Consequently, the ECWMF data was chosen and subsequently utilized in the calculations.

## 2.3 Remote sensing measurements

### 2.3.1 CAELI Raman lidar

CAELI is a high power multiwavelength Raman lidar system that is specifically designed for profiling water vapor, aerosols, and clouds (Apituley et al., 2009). CAELI uses a pulsed neodymium-doped yttrium aluminium garnet (Nd:YAG) laser as the light source, emitting laser pulses at 1064 nm (IR), 532 nm (VIS), and 355 nm (UV). The laser and receiver are aligned in a dual-axis configuration with a single target axis pointing vertically to the zenith. The receiving system uses Newtonian telescopes and separate optical channels, with three elastic channels (1064, 532, and 355 nm) and three Raman channels (387 and 607 nm (nitrogen) and 407 nm (water vapor)) to detect the backscattered light signals in the atmosphere. For full tropospheric coverage, CAELI's receiving system is duplicated using a 15 and a 57 cm telescopes for near field range (NFR) and far field range (FFR) measurements respectively. More details on CAELI can be found in Apituley et al. (2009). For this study, the lidar aerosol optical products were retrieved using the EARLINET Single Calculus Chain (SCC) using CAELI's near field telescope measurements and atmospheric model data (D'Amico et al., 2015). To increase signal-to-noise ratio, the raw data vertical resolution was reduced to 60 m and the profiles were usually accumulated for about 1 hour. The Raman backscatter profiles are available starting from 150 m above ground, while the elastic backscatter and Raman extinction coefficient were retrieved above 810 m. To avoid the effects of an incomplete overlap on the extinction retrievals, a post-

---

[5] https://cloudnet.fmi.fi/ (last access 20 July 2022)
[6] https://dataplatform.knmi.nl(last access: 20 July 2022)



processing step consisted in removing the extinction profile below the full overlap height (690 m for a telecover test deviation < 3%) plus the effective vertical resolution of the extinction retrieval (Mattis et al., 2016), which resulted in a minimum usable Raman extinction profile typically above 810-1050 m depending on the aerosol load, solar background, and fraction of cloud screened profiles.

### 2.3.2 CHM15k Ceilometer

The CHM15k ceilometer is a single-wavelength elastic-backscatter lidar manufactured by Lufft (2019), Germany. The CHM15k employs a Nd:YAG narrow-beam microchip laser that emits 1 ns pulses at a wavelength of 1064 nm, with a repetition rate ranging between 5-7 kHz and a receiver field of view of 450 µrad. The laser sensor is capable of measuring heights up to 15 km, with an initial overlap point of 80 m and complete overlap achieved at 800 m above ground (Hervo et al., 2016; Brunamonti et al., 2021). Wiegner and Geiss (2012) reported a relative error of 10% in backscatter coefficient at 1064 nm retrieved through this methodology using a similar system (CHM15kx by Jenoptik, Germany). The data used in this study were processed by the Eumetnet E-PROFILE ALC data hub: https://www.eumetnet.eu/activities/observations-programme/current-activities/e-profile/. The calibrated data with a vertical resolution of 30 m, and a time resolution of 5 minutes can be requested from the KNMI Data Platform (https://dataplatform.knmi.nl/). The ceilometer data was primarily used to aid in the visual discrimination between lofted aerosol layers (possibly from long range transport) and the boundary layer aerosols from recent mixing processes. An overview of the CHM15k measurement with the marked CAELI Raman lidar measurements times (from May 2021 to October 2021) is given in Figure1.

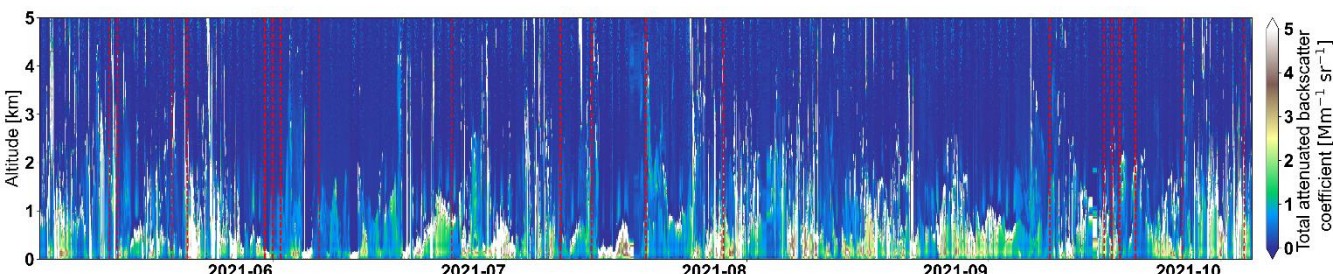

**Figure 1: Backscatter coefficient at 1064 nm of CHMk15 ceilometer measurements at CESAR site from May to November in 2021. The red dash lines represent the CAELI measurements availabilities.**



## 2.4    Calculations

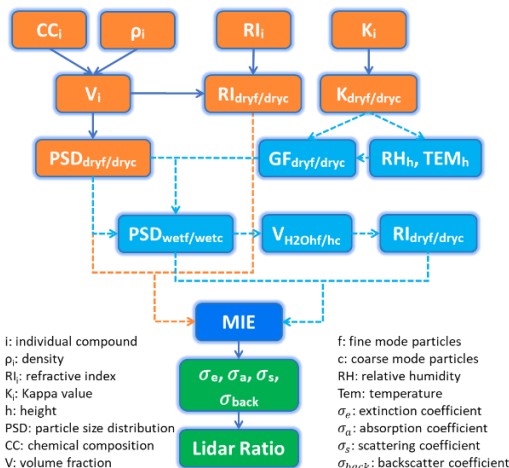

**Figure 2: The flow diagram of the calculations**

The aerosol optical properties (including the scattering coefficient, absorption coefficient, extinction coefficient, backscatter
coefficient, and lidar ratio) were calculated based on Mie theory as displayed in the Figure 2. The main measurement data
inputs were: (i) Chemical composition measured by the ACSM and MAAP as described in the section 2.1.1. (ii.) PSD measured
by the MPSS and APS as described in the section 2.1.2. (iii.) RH and temperature profiles obtained from the ECMWF as
described in the section 2.2.

Starting with the dry aerosol, the PSD data was separated into fine mode (< 2.5 µm) and coarse mode (> 2.5µm). We assumed
that the fine mode was composed of an internal mixture of secondary inorganic aerosol (SIA) including ammonium, nitrate
and sulphate, and organics, and that BC was externally mixed. In addition, we assumed that the coarse mode was composed
of sea salt (SS) and mineral dust (MD) (Schaap et al., 2010). Because the coarse mode chemical composition was not measured
during the RITA campaign, we employed the average SS and MD fractions obtained from the previous Trolix campaign in
2019[7], which indicated a composition of 70% SS and 30% MD in volume fraction. The used densities of the SS and MD are
listed in Table S3 and calculation details are in the supplementary materials section S8. In addition, we conducted a sensitivity
analysis by considering two extreme scenarios: one where the coarse mode was entirely composed of SS and another where it
was entirely composed of MD. The outcomes of these sensitivity tests will be elaborated upon in the subsequent discussion.
A uniform chemical composition was assumed for fine and coarse mode, respectively.

The refractive index (RI) of the fine and coarse mode was calculated as volume-weighted average of the RI of the individual
constituents (as shown Table S3). The volume of each species was calculated as the measured mass concentration divided by
the corresponding density, also shown in Table S3. Given the PSD, as well as the RI of the dry aerosols, a Mie model
(PyMieSca v1.7.5; (Sumlin et al., 2018)) was used to calculate the optical properties, namely the aerosol scattering coefficient,

---

[7] https://ruisdael-observatory.nl/trolix19-tropomi-validation-experiment-2019/



backscatter coefficient, extinction coefficient and lidar ratio at the wavelengths of the Nephelometer. The calculations were compared to the measured scattering coefficient and backscatter coefficient as discussed in section 3.1.

For the ambient aerosol, we followed the same strategy to separate the fine mode and coarse mode as described for the dry aerosols, i.e. separate growth factors are derived for fine (only for SIA because BC was considered as non-hygroscopic) and coarse mode. The ambient PSD was calculated by multiplying the dry particle diameters with a diameter growth factor (GF) derived for the respective RH and temperature at different height above ground. Specifically, the GF for a given saturation ratio S (Gf(S)) is estimated based on kappa values (Zhang et al., 2017; Zou et al., 2019; Petters and Kreidenweis, 2007).

$$GF(S) = \left(\frac{K_{mix}S}{K_e - S} + 1\right)^{1/3},$$
(1)

$$K_e = \exp\left(\frac{4\sigma M_w}{RT\rho}\right),$$
(2)

where the $K_{mix}$ is the volume weighted average of the individual kappa values of the compound classes listed in the Table S3. For SIA, upper and lower limits (0.5-0.7) were used and accounted for in the uncertainty of the calculated optical properties. $K_e$ is essentially a constant at a fixed temperature, where $\sigma$ is the surface tension of the solution/air interface ($\sigma = 0.072$ J m$^{-2}$),

$M_w$ is the molecular weight of water ($M_w = 18$ g mol-1), R is the universal gas constant (R = 8.3145 J mol$^{-1}$ k$^{-1}$), T (K) is temperature, $\rho$ is the density of water ($\rho = 1000$ kg m$^{-3}$). With the GF, the total water volume concentration can be obtained from the difference between the wet integral particle volume size distribution and the dry integral particle volume size distribution based on the following equation:

$$V_{H2O} = \sum_i \frac{\pi D_{dryi}^3}{6}(GF^3 - 1)dn_i,$$
(3)

where the $dn_i$ is the number concentration (cm$^{-3}$) of size bin i and $D_{dry}$ is the dry particle diameter (nm). The wet RI was calculated as the volume weighted average of the individual RIs of all chemical constituents, now including the calculated water volume concentration in addition to the original volume concentrations. Finally, the optical properties of the ambient aerosol were calculated based on the Mie model with ambient PSD and RI as input parameters. The vertical profiles of the optical properties were derived by using the corresponding meteorological profile with the assumption of a homogenous

distribution of the aerosol within the boundary layer height. The vertical profiles predicted by the model were compared with the Raman lidar measurements in the following section 3.2.

We quantified calculation uncertainties by assessing a set of nine parallel experimental results. These results were obtained through cross-testing, specifically by varying two key parameters as mentioned in the previous content: the volume fraction of SS and MD with values of 1, 0.7, and 0; and the SIA kappa values of 0.5, 0.6, and 0.7. The standard deviation of these parallel

results serves as calculation uncertainties.



# 3    Results and discussion

## 3.1    Optical properties compared by calculation and Nephelometer at ground level

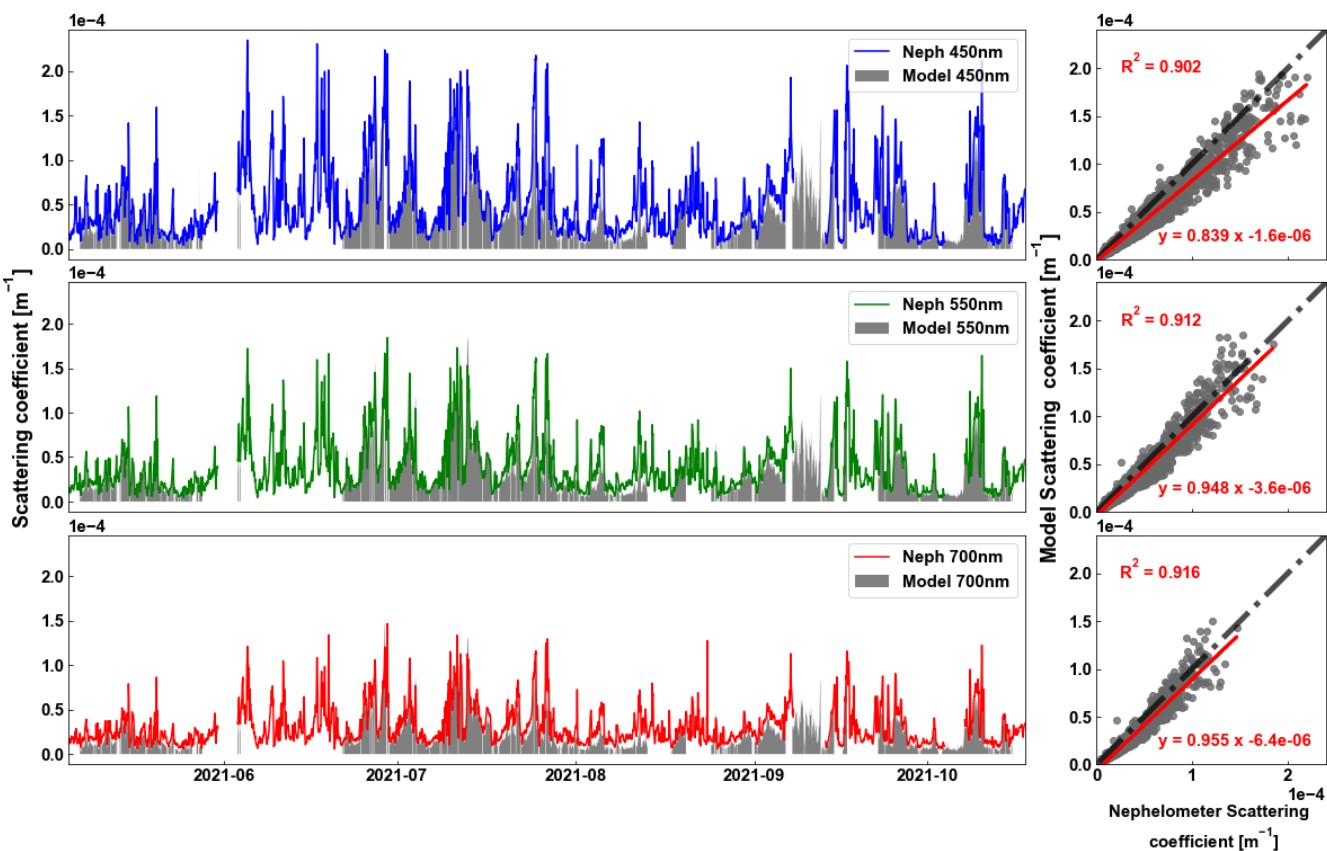

**Figure 3: Time series of the scattering coefficient at 3 wavelengths (450nm, 550nm 700nm from the top to bottom, respectively)**
**255    measured by the Nephelometer (coloured lines) and calculated from the Mie model (grey shades) in the left panel; a scatter plot of**
**each wavelength between the measured scattering coefficient (horizontal axis) and the calculated scattering coefficient (vertical axis)**
**in the right panel. The red line represents the regression line, and the black dashed line represents the 1:1 line.**





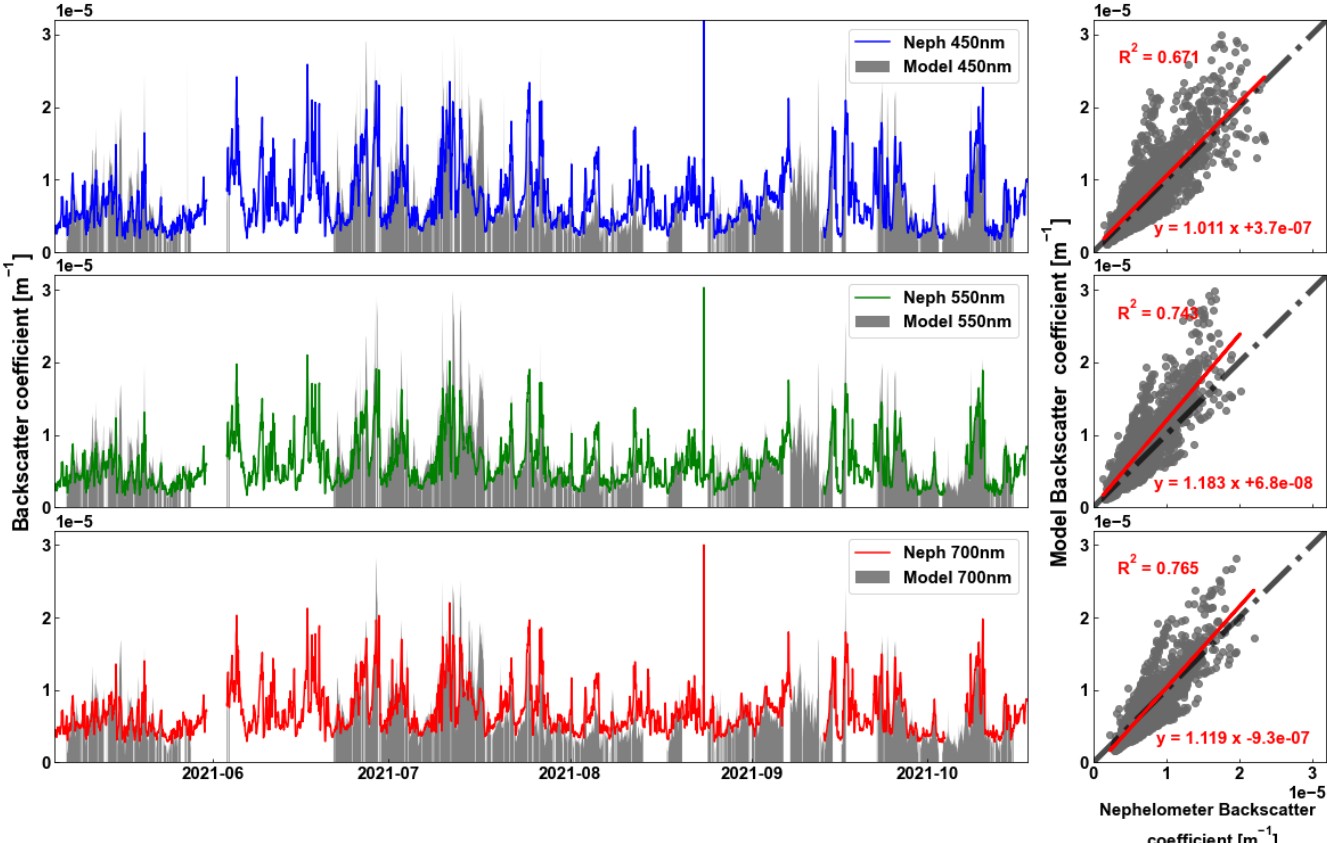

**Figure 4: Time series of the backscatter coefficient at 3 wavelengths (450nm, 550nm 700nm from the top to bottom, respectively)**
**measured by the Nephelometer (coloured lines) and calculated from the Mie model (grey shades) in the left panel; a scatter plot of each wavelength between the measured backscatter coefficient (horizontal axis) and the calculated scattering coefficient (vertical axis) in the right panel. The red line represents the regression line, and the black dashed line represents the 1:1 line.**

The Nephelometer (at 450 nm, 550 nm, 700 nm) was operated continuously for measuring the aerosol scattering coefficient and backscatter coefficient at RH below 40%. Data from May to the end of October during the RITA-2021 campaign were
used to validate the model calculations. Figure 3 and Figure 4 show the time series of the scattering coefficient and backscatter coefficient at 3 wavelengths obtained by Nephelometer measurements and the calculations outlined in section 2.4. The corresponding scatter plots including best fit lines are given on the right. Gaps in the calculated data are mainly due to maintenance and power failures of the aerosol in situ instruments, but the data coverage is more than 90%. Good agreement was found between the measured and calculated scattering coefficients, with a slope of 0.84 ($R^2 = 0.90$) for 450 nm, and 0.95
($R^2 = 0.91$) for 550 nm, as well as 0.96 ($R^2 = 0.92$) for 700 nm. The model slightly underestimated the measurements, but the difference becomes smaller at larger wavelengths. Good agreement was also found for the backscatter coefficient, with the slope of the calculated values vs the measured values given as 1.01 ($R^2 = 0.67$) for 450 nm, and 1.18 ($R^2 = 0.74$) for 550 nm, as well as 1.12 ($R^2 = 0.77$) for 700 nm. The model calculations shown in the Figure 3 and Figure 4 assume that the coarse mode is composed of 70% SS and 30% MD as described in section 2.4. Results from a sensitivity study assuming that the coarse

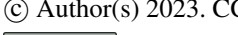



mode consisted either entirely of SS or entirely of MD are presented in Figure S6 and Figure S7, and are very similar to the results in Figure 3 and Figure 4. More specifically, under the given particle size distribution conditions, the scattering coefficients for these two extreme scenarios differ on average less than 4% across varying wavelengths, the backscatter coefficients less than 19%. This shows that the backscatter coefficient is more sensitive to the coarse mode chemical composition, which can explain the lower $R^2$ values in Figure 4 compared to Figure 3. However, in general an average chemical

composition of the coarse mode for the site is sufficient to predict the optical properties with reasonable accuracy. This is a considerable advantage, as the coarse-mode chemical composition is usually not as readily available as the fine mode composition for many sites. For sites, where the coarse mode comprises a very high mass fraction of PM10, a more accurate representation of the coarse mode chemical composition might be necessary to predict the backscatter coefficient.

### 3.2     Comparison between the modelling and Raman lidar

The time periods when the CAELI Raman lidar was operated are marked in Figure 1. Due to unsuitable weather conditions, e.g. shallow atmospheric boundary layer or low clouds layers, it was not possible to retrieve lidar profiles for all time periods. Table S1 summarizes which in-situ data were available on the dates of the Raman lidar measurements. Four representative examples (two polluted cases and two clean cases) were selected and are discussed in detail in the following section. The rest of the profiles are displayed in the supplementary material section 6.




### 3.2.1 Polluted cases

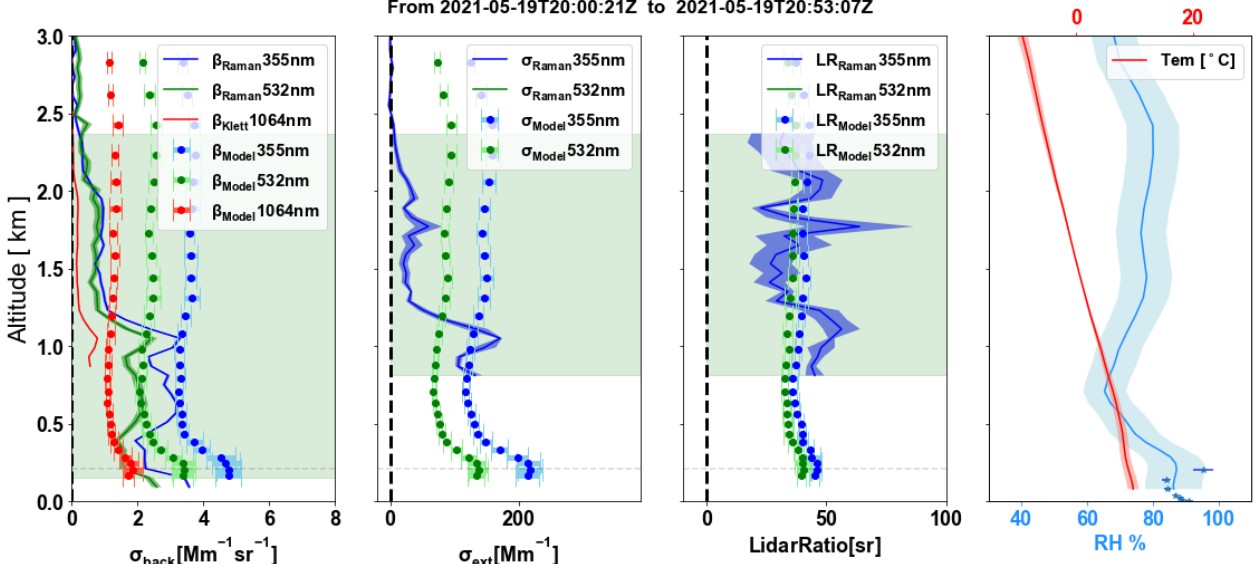

**Figure 5: Vertical profiles of the aerosol optical properties (a) the Raman lidar backscatter coefficients ($\sigma_{back}$) at 355 nm (blue line), 532 nm (green line), and 1064 nm (using the Klett method; red line). The uncertainties of the measurements are given by the shaded areas. The predicted backscatter coefficient given by the Mie model at 355 nm (blue dots), 532 nm (green dots), and 1064 nm (red dots) with error bars representing the model uncertainties. (b) the corresponding extinction coefficient ($\sigma_{ext}$) profiles of the lidar measurements and model calculations. (c) the corresponding retrieved and calculated lidar ratio profiles. The light green background represents upper and lower limits of the valid lidar measurements. The grey dashed horizontal line represents the lowest layer appearance. (d) The vertical profiles of the meteorological data with RH in blue, and temperature in red, as well as the in situ measured RH (blue stars). Data from 20:00:21 to 20:53:07 at UTC time on May-19-2021.**

Figure 5 shows the averaged vertical profiles of the aerosol optical properties retrieved from the Raman lidar during the period of 20:00:21 to 20:53:07 (UTC) on May-19-2021. The 10-min ground measurements were averaged to best represent the lidar start and stop times, in this particular case from 20:00 to 20:50. The meteorological profiles were given by the ECMWF model with 1-hour time resolution with an uncertainty of 10%. The uncertainties provided in the model calculations correspond to the standard deviation of the various sensitivity studies as explained in section 2.4. The valid lidar measurement levels are marked in the Figure 5. The lowest altitude for the backscatter coefficient is above 150 m, whereas the lowest altitude for the extinction coefficient and lidar ratio is determined by the valid retrievable range as described in section 2.3.1. Furthermore, the upper limits were manually selected to only include aerosols originating from the planetary boundary layer (including the residual layer, if present) for all the profiles, excluding lofted layers possibly originating from long-range transport. All the subsequent profiles adhered to the same approach. In this case, the dataset spanning from 810 m to 2370 m was employed for subsequent lidar ratio calculations and comparison to the model calculations.

For this study case, several aerosol layers were observed approximately below 1000 m, which can be seen in the Raman lidar image (in Figure S9). For the backscatter coefficient profiles, the simulated values are higher than the measured values below 500 m, which is likely due to inaccuracies in RH of the ECMWF data. This is highlighted by discrepancy between ECMWF



data and situ RH measurements at 200 m as shown in the Figure 5(d). Between the altitudes of 600 m and 1100 m the backscatter coefficient does not change strongly with altitude and retrieved and calculated backscatter coefficients agreed within 12%. Specifically, within this range, the lidar reports values of 2.9 Mm$^{-1}$ sr$^{-1}$ for 355 nm and 2.0 Mm$^{-1}$ sr$^{-1}$ for 532 nm, comparable with calculated values of 3.3 Mm$^{-1}$ sr$^{-1}$ for 355 nm and 2.1 Mm$^{-1}$ sr$^{-1}$ for 532 nm. Beyond 1100 m, the measured backscatter coefficients rapidly decrease to nearly 0 above the mixed layer and the comparison with ground-based data ceases to be meaningful. The Raman lidar image also shows increased backscatter values at around 1000 m, potentially indicating a different aerosol type or significant RH changes, which might not be well captured by the ECMWF data.

For the extinction coefficient profiles, the Raman lidar retrievals provided good quality data only for 355 nm. A limited overlap existed between the valid lowest retrieved level and the aerosol layer at 1000 m, posing challenges for direct comparisons. Nevertheless, the average extinction coefficient from 800 m to 1100 m is approximately 110 Mm$^{-1}$ at 355 nm for both lidar measurements and the calculations.

Finally, the retrieved lidar ratio is $39.7 \pm 10.6$ sr$^{-1}$ at 355 nm for the valid altitudes, whereas the calculations yield a lidar ratio of $40.1 \pm 1.6$ sr$^{-1}$ at 355 nm and $35.3 \pm 1.4$ sr$^{-1}$ at 532 nm. This range of values is typical for a polluted aerosol type (Bohlmann et al., 2018; Groß et al., 2013; Illingworth et al., 2015). This classification is in line with ACSM measurements (refer to Figure S10), which indicate an average non-refractory PM$_{2.5}$ mass concentration of $10.01 \pm 0.23$ µg m$^{-3}$. Notably, nitrate (50.9%) and ammonium (17.4%) contribute significantly to this mass concentration. Moreover, as shown in Figure S11, the analysis of back trajectory using the Hysplit model (Stein et al., 2015; Rolph et al., 2017) implies that the air masses originated from the sea, but were transported over Ireland and the United Kingdome and also the northwest of the Netherlands, resulting in elevated levels of anthropogenic pollutants.

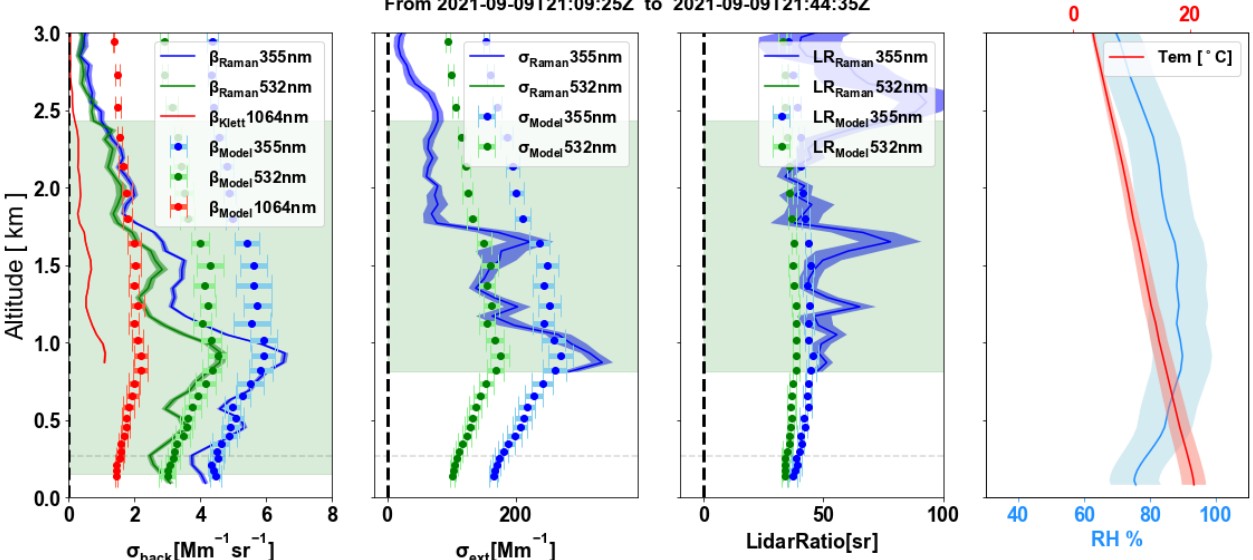

Figure 6: Corresponding to Figure 5 for the period from 21:09:25 to 21:44:35 at UTC time on Sep-09-2021.



Figure 6 shows a second polluted period from 21:09:25 to 21:44:35 (UTC) on Sep-09-2021. The lowest layer was found around 270 m. The valid retrieval range is from 810 m to 2430 m. The Raman lidar images displayed in Figure S12 shows a complex and variable cloud structures during this period. The profiles of backscatter coefficients obtained from Raman lidar retrievals and calculations agree remarkably well from the surface up to an altitude of 1000 m within the mixed layer height. On average,

the differences between the two datasets are less than 5% for both 355 nm and 532 nm. Additionally, the backscatter coefficient profiles increased from the surface to 1000 m, from 3.9 to 6.5 $Mm^{-1}$ $Sr^{-1}$ for 355 nm and from 2.7 to 4.7 $Mm^{-1}$ $Sr^{-1}$ for 532 nm. This increase is reflected in the RH profile (from 75% to 90%). The variations in aerosol optical properties within 1 km altitude were thus primarily due to changes in relative humidity and could be well predicted by ground-based data. However, the ground-based aerosol information is no longer applicable to profiles situated above 1 km.

The extinction coefficient profiles exhibit higher values initially at around 850 m, followed by a rapid decrease up to an altitude of approximately 1.2 km. The calculated extinction coefficient also decreases above 850 m but much less. This could be partly due to the coarse resolution of the ECMWF RH profiles, but most likely results from a lower aerosol concentration above the mixed layer. Nevertheless, the profiles of the measurements and calculations at 355 nm agreed reasonably well in the altitude range of 900 m to 1000 m. Finally, the retrieved and calculated lidar ratios, which do not depend on the absolute aerosol

concentration closely agree throughout the column, indicating a same aerosol type. The lidar average retrievals over the valid retrieval height yielded a value of 49.1 ± 10.1 $sr^{-1}$, while the model calculations produced a value of 43.2 ± 1.7 $sr^{-1}$ at 355 nm. Similar to the lowest polluted case, the ACSM measurements showed a $PM_{2.5}$ mass concentration of 12.61 ± 0.63 µg $m^{-3}$, primarily dominated by organic components (43%), as illustrated in Figure S13. Furthermore, the analysis of back trajectory, as depicted in Figure S14, demonstrated that the air masses originated from southern Europe.






### 3.2.2    Clean cases

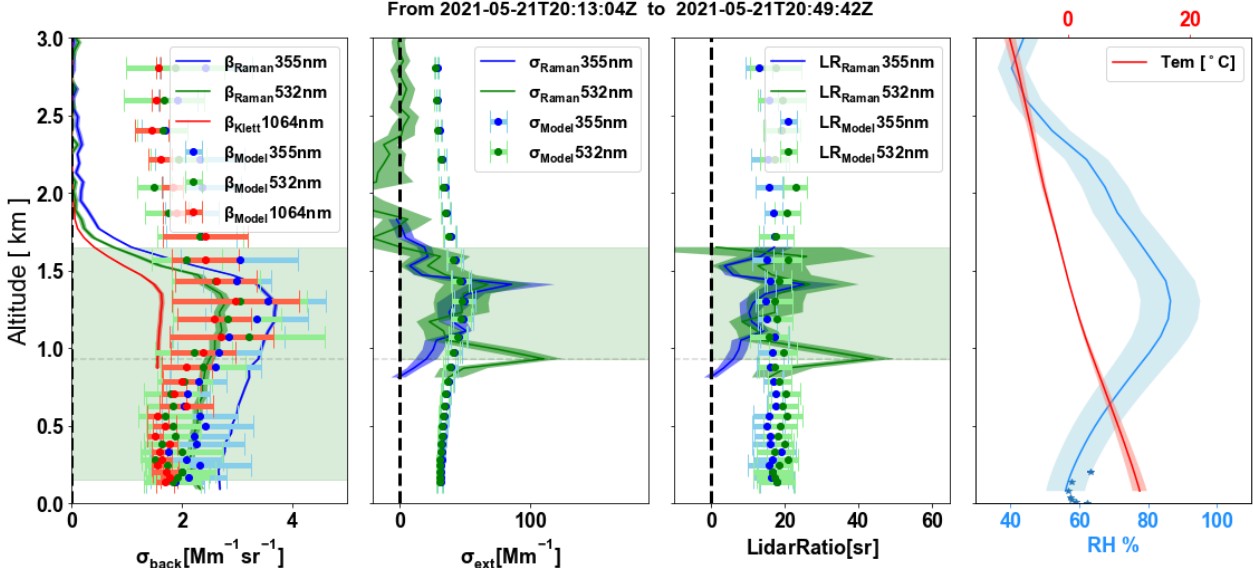

**Figure 7: Corresponding to Figure 5 for the period from 20:13:04 to 20:49:42 at UTC time on May-21-2021.**

Figure 7 shows the vertical profiles of the aerosol optical properties retrieved from the CAELI Raman lidar during the period

of 20:13:04 to 20:49:42 (UTC) on May-21-2021. During the observed period, the PBLH was identified at an altitude of approximately 1650 m, with the lowermost layer of valid extinction coefficient retrieval extending to around 930 m, as indicated in Figure S15. Within this altitude range, a good agreement is observed between retrievals and calculations for the backscatter coefficients at both 355 nm and 532 nm. Moreover, the retrievals illustrate a gradual increase in backscatter coefficients from the surface up to approximately 1.4 km altitude. Specifically, at 355 nm, the backscatter coefficient increases

from approximately 2.7 Mm$^{-1}$ sr$^{-1}$ to ~3.4 Mm$^{-1}$ sr$^{-1}$ for 355 nm and from ~2.3 Mm$^{-1}$ sr$^{-1}$ to ~2.4 Mm$^{-1}$ sr$^{-1}$ for 532 nm as the RH increases from around 50% at the surface to about 85% near the top of the PBLH. A similar increase in the calculated backscatter coefficient indicates that this increase is predominantly driven by aerosol hygroscopic growth. However, the calculated backscatter coefficient below 1 km was slightly lower than the retrievals, which might be due to the uncertainties in the RH profile. This is supported by Figure 7(d), where in situ RH measurements show a more pronounced RH increase

above 200 m, whereas ECMWF observations increase less drastically. Furthermore, significant uncertainties are evident in the model-derived estimates. In this case, the coarse mode accounts for approximately 44% ± 6% of the total backscatter coefficient on average, and the assumption of the coarse mode being either pure SS or pure MD causes large uncertainties.

The Raman lidar extinction coefficient and lidar ratio exhibit an anomalous spike at the onset of the valid retrieval height, which will be disregarded in the subsequent analysis and discussions. Within the altitude range of 1 km to 1.6 km, the model

calculations closely matched the retrieved extinction coefficient on average. Notably, the profile of extinction coefficient at 355 nm and 532 nm showed a broad peak in the region of maximum RH, which was also captured by the model, further indicating that the variation with altitude was mainly caused by aerosol hygroscopicity growth. The average retrieved lidar





ratios are 11.5 ± 5.8 sr$^{-1}$ for 355 nm and 18.1 ± 6.6 sr$^{-1}$ for 532 nm, and the average calculated lidar ratios are 15.8 ± 1.0 sr$^{-1}$ for

355 nm and 18.3 ± 1.8 sr$^{-1}$ for 532 nm. These lidar ratios are typical for marine aerosols (around 5 to 30 sr$^{-1}$) (Bohlmann et al.,

2018; Illingworth et al., 2015; Groß et al., 2013). The ACSM and MAAP measurements showed a very low PM$_{2.5}$ mass

concentration with an average of 0.89 ± 0.07 μg m$^{-3}$ during this period, and relative mass fractions of 27% organic aerosol,

followed by 25.0% BC, 21.6% sulphate, 11.3% nitrate, 9.9% ammonium and 5.1% chloride as shown in Figure S16. While

the ACSM is unable to measure NaCl, the presence of higher levels of chloride ions (below 1 % on average during the

measurements) may suggest the existence of other chlorine-rich, inorganic salt particles. The back trajectory analysis as shown

in Figure S17, further proved that the air masses originated from the sea with very clean background aerosols.

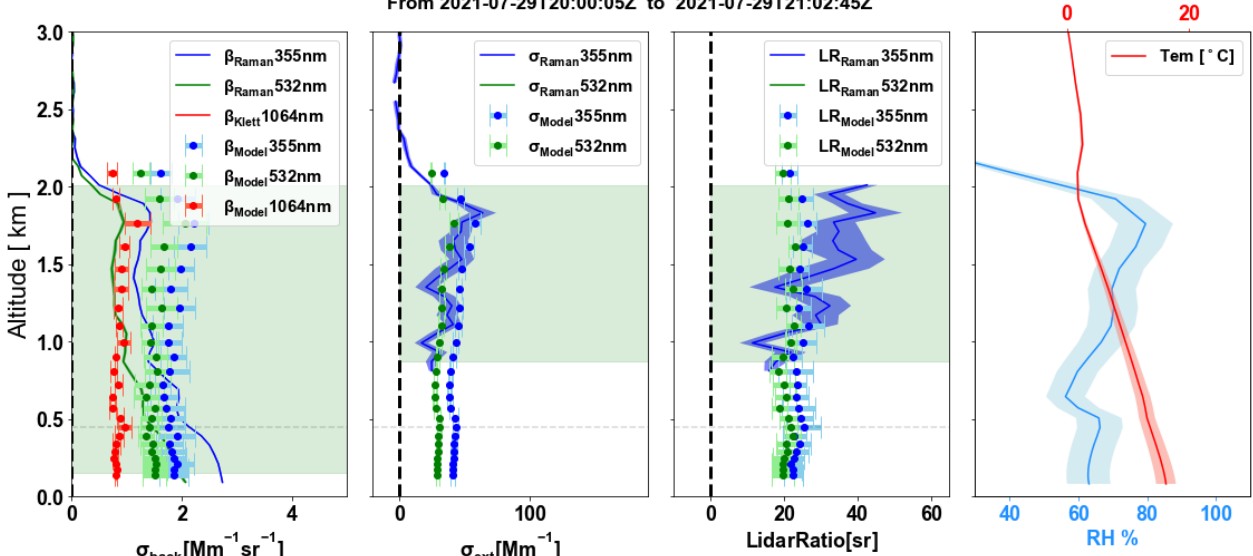

**Figure 8: Corresponding to Figure 5 for the period from 22:00:05 to 21:02:45 at UTC time on July-29-2021.**

Figure 8 shows the profiles for the period from the 20:00:05 to 21:02:45 (UTC) on July-29-2021. The lowermost layer is

situated at an altitude of approximately 450 m and the applicable range for the extinction profile and lidar ratio spans from 870

m to 2010 m. The retrieved backscatter coefficient decreases with altitude, but the calculated backscatter coefficient is rather

constant with altitude. The calculations underestimate the retrievals at altitudes below 500 m and overestimate the retrievals

(by approximately 20% - 30%) at altitudes around 1500 m. This difference can be attributed to (i) variations in aerosol

concentrations or chemical properties between ground-level and higher altitudes; (ii) a potentially inaccurate RH profile near

the ground, where the 1hr time resolution of the re-analysis data do not capture correctly the development of a nocturnal stable

layer near the ground; (iii) other inaccuracies in the model such as insufficient information on the size resolved chemical

composition and aerosol mixing state; (iv) on the other hand, the lidar retrievals near the ground could also be inaccurate,

especially at the low aerosol concentrations in this clean case. However, despite these discrepancies, the model provided values

are on the order of magnitude of the retrievals and agree within uncertainties for a large part of the profile.



The agreement between the modelled and retrieved values of extinction coefficient and lidar ratio profiles within the altitude
range of 900 m to 1800 m suggests a reasonable representation of aerosol properties by the ground-based measurements throughout the boundary layer. Especially, the lidar ratio obtained from the Raman lidar measurements at 355 nm was determined to be $30 \pm 8.3$ $sr^{-1}$, while the model estimated values were $25.0 \pm 1.3$ $sr^{-1}$ at 355 nm and $21.5 \pm 1.1$ at 532 nm. These lidar ratio values are consistent with aerosols originating from marine sources during the observed period, which is supported by the back trajectory shown in Figure S20. Additionally, the low aerosol mass concentration ($2.41 \pm 0.51$ µg $m^{-3}$) shown in
Figure S19 with a significantly higher fraction of sulphate (45.7%) further supports this result.

There are 22 more profiles displayed in the supplementary material section 6, where it shows extinction and backscatter coefficients are sometimes severely under- or overestimated by the ground-based calculations. However, the lidar ratios are much better predicted. We speculate that the main reasons for this phenomenon are as follows: Although upper-level aerosols may have similar chemical composition and size distribution as surface-level aerosols, they may be present at different
concentrations. Thus, the backscatter or extinction coefficients of aerosols may be overestimated or underestimated by the same factor resulting in a similar lidar ratio. Or it could be the meteorological data may not be sufficiently accurate. Especially when relative humidity is overestimated or underestimated, this has a more significant influence on extinction and backscatter coefficients, but its impact on the lidar ratio is less pronounced. Another crucial factor may be the influence of shape effects, which normally become more significant for the larger particles. Previous studies show that the backscatter cross-section and
extinction cross-section may be underestimated or overestimated by a factor ranging from -2 to +5, depending on the particle shapes and size ranges (Potenza et al., 2016; Geisinger et al., 2017).

## 3.3    Summary of the lidar ratio comparison

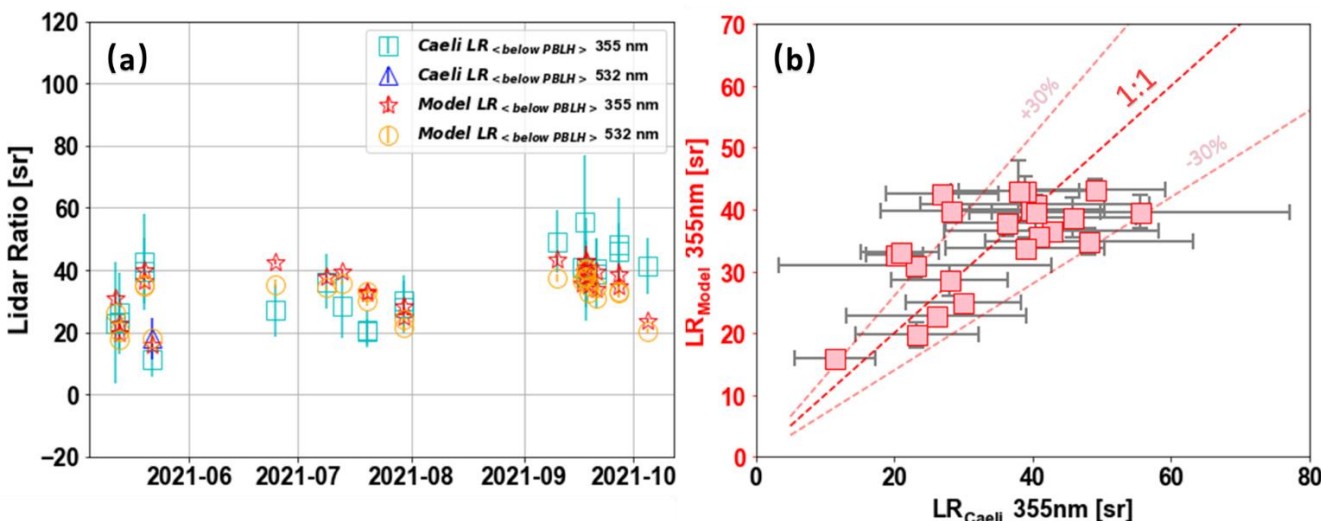

**Figure 9: (a) Time series of the mean lidar ratio and its uncertainty (at 355nm and 532 nm) from the valid lidar retrievals and model**
**calculations. (b) Scatter plot of the lidar ratios (LR) from Raman lidar measurements (x axis) and from calculations (y axis) at 355**
**nm.**





Figure 9 (a) shows the time series of the average lidar ratios (at 355nm and 532nm) for each period retrieved from lidar measurements and calculated by the model for the corresponding valid retrieval levels of each profile. The error bars correspond to the standard deviation of each effective lidar ratio profile. For the most part, the simulated lidar ratios are comparable to the lidar ratios measured by Raman lidar. Significant standard deviations in Figure 9 (a) illustrate that the measured lidar ratios were quite variable across the planetary boundary layer, which might indicate different aerosol layers at various altitudes. This cannot be taken into account for the calculations. Consequently, the model-generated lidar ratios tend to remain relatively stable with altitude for the majority of cases. Nevertheless, the simulated lidar ratios are within the range of retrieved lidar ratios, with differences from case to case are usually smaller than ± 30% for the wavelength at 355nm as shown in Figure 9 (b). On the whole, the calculated lidar ratios were in the range of 16 - 43 sr at 355 nm and 18 - 41 sr at 532 nm on average, indicating a relatively low-pollution environment. Furthermore, the model calculations show that the lidar ratio has a small wavelength dependence, with on average higher lidar ratio at 355 nm (slope of 0.64 and $R^2$ of 0.94 between the lidar ratio at 355 nm and 532 nm). This is consistent with findings from Mattis et al (2004), which summarized a long-term Raman lidar measurements with the lidar ratio from 2000 to 2003 for central European haze, specifically the anthropogenic aerosol particles, with values of 58 ± 12 sr for 355 nm, 53 ± 11 sr for 532 nm, and 45 ± 15 sr for 1064 nm wavelengths in the upper part of the PBL. In the free tropospheric and stratospheric layers, the lidar ratio possibly has a different wavelength dependence, where Moritzet et al. (2004) reported that the lidar ratios were 40 - 45 sr for 355 nm, 65 - 80 sr for 532 nm, and 80 - 95 sr for 1064 nm.

In summary, through the combination of ground-based aerosol measurements with the readily accessible ECMWF data, we can predict aerosol optical properties within the mixed layer relatively well. In particular, the lidar ratio can be predicted throughout the boundary layer in the absence of strong elevated aerosol layers, such as dust layers. However, it is crucial for the model to be furnished with an accurate measured particle size distribution including the coarse mode. While possessing chemical insights into the coarse mode is an advantage, it is not absolutely necessary. However, the uncertainties of the calculations might be significant in regions the coarse mode particles are dominant. Within the mixed layer, our results show that the enhancement of the backscatter coefficient and extinction coefficient is mainly driven by hygroscopicity growth. Consequently, the availability of accurate and high vertical resolution RH profile is important for constructing a robust model input, but even one-hourly ECMWF humidity fields give reasonable results.

## 4    Conclusions

In this study, a Mie theory-based model was applied to ground-based in-situ measurements to predict ambient aerosol optical properties including scattering coefficient, backscatter coefficient, extinction coefficient and lidar ratio. The input data are: (i) aerosol chemical composition and (ii) particle size distribution measured at the surface; (iii) the meteorological data from European Centre for Medium-Range Weather Forecasts (ECMWF). The data was collected during the Ruisdael land-atmosphere interactions Intensive Trace-gas and Aerosol (RITA) 2021 campaign, at the CESAR site in the Netherlands with

a total time span of 5 months (from May to October). The calculations were first validated by comparing to observations from
TSI integrating nephelometer at dry conditions for the entire period. The calculations and measurements across multiple
wavelengths with slopes of 0.84 - 0.96 ($R^2 \geq 0.90$) for the scattering coefficients, and slopes of 1.01 – 1.18 ($R^2 \geq 0.67$) for the
backscatter coefficients. Furthermore, the model was compared with aerosol optical vertical profiles retrieved by a multi-
wavelength Raman lidar. The results showed that, for a homogeneously distributed aerosol, the model could effectively
simulate the vertical profile of the aerosol backscatter coefficient as a function of relative humidity within the mixed layer.
The comparison of extinction coefficients posed challenges due to the limited overlap between the lower layer of retrievals
and the mixed layer. However, the profiles at the shared levels exhibited a reasonable connection, suggesting a meaningful
comparison could still be made. The simulated lidar ratio can predict the measured lidar ratio within ± 30% for the average
values below the planetary boundary layer height. Overall, our study shows that, besides the particle size distribution and
chemical composition, the relativity humidity is a crucial input for the model to generate accurate backscatter and extinction
coefficient profiles. Moreover, in the boundary layer, it is usually possible to approximate the lidar ratio using ground-based
measurements. This approach allows for the extension of extinction profiles to lower altitudes that are typically challenging to
retrieve, or it can be employed alongside basic backscatter lidar systems to calculate the extinction and then further the aerosol
optical depth, which could potentially extend to forecasting aerosol optical depth and could offer advantages in extensive-scale
or worldwide radiation simulations.

**Data availability**

The most data involved in this study is part of the Ruisdael Observatory (https://ruisdael-observatory.nl) project. The ground
based measurements can be accessed at repository under https://doi.org/10.5281/zenodo.7924288 (Liu et al., 2023). The in situ
meteorological data and Ceilometer data are available at the KNMI Data Platform (https://dataplatform.knmi.nl). Other remote
sensing data can be accessed from the authors upon reasonable request.

**Author contribution**

XL, BH, and UD designed this study. DG, AA, AH, DD, UD and XL implemented the experiment and sample analysis. DG
and AA provided the LIDAR retrievals. XL analysed the data and wrote the manuscript. All co-authors proofread and
commented on the paper.

**Competing interests**

The authors declare that they have no conflict of interest.



**Acknowledgements**

The Chinese Scholarship Council (No.201906350118) is acknowledged for the financial support for the author X. Liu. In the project we make use of the Ruisdael observatory infrastructure, funded by the Dutch Science foundation NWO (grant number 184.034.015). The authors thank Delft University of Technology for providing data of the microwave radiometer, which is also an instrument of the Ruisdael Observatory.

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
