# Peer review of "Aerosol optical properties within the atmospheric boundary layer predicted from ground-based observations compared to Raman lidar retrievals during RITA-2021"

_EGUsphere, 2023_

## Author Comment (AC1)

Response to referees for manuscript "Aerosol optical properties within the atmospheric boundary layer predicted from ground-based observations compared to Raman lidar retrievals during RITA-202" by Xinya Liu, Diego Alves Gouveia, Bas Henzing, Arnoud Apituley, Arjan Hensen, Danielle van Dinther, Rujin Huang, and Ulrike Dusek. (manuscript ID: EGUSPHERE-2023-2262)

Referee #2 comments:

The authors present a comparison of backscatter and extinction coefficients as inferred from ground-based in-situ measurements and lidar observations. The manuscripts describe the steps taken to convert in-situ measurements to optical properties and shows findings for a five-month measurement campaign in 2021. Given these specifics, the manuscript would potentially qualify as a Measurement Report. However, I don't find the work to be within the scope of ACP and there are several issues that lead me to recommend rejection:

1. The authors fail to motivate why the scientific community should be interested in this work. Will the method be applied in future analyses of CAELI observations? Can it be adapted to other sites? How is it superior to the traditional approach of just assuming a lidar ratio?

In response to the referee's feedback regarding the motivation behind our work, we have taken steps to more clearly state the significance and relevance of our study in the revised manuscript, particularly in the abstract. Here, we wish to emphasize several key points why such a study is interesting and worthwhile:

a. **Novelty:** The methodology developed in this study provides a valuable approach for obtaining vertical profiles of the lidar ratio through in situ measurements and readily accessible ECMWF meteorological data. This is particularly valuable for atmospheric research sites lacking advanced lidar techniques. We show that for realistic aerosol size distributions the lidar ratio is rather insensitive to relative humidity and can be used to evaluate typical lidar ratio assumptions more quantitatively.

b. **Integration**: Our research contributes significantly to bridging the gap between in situ measurements and remote sensing lidar observations. The methodology provides a good connection from the ground to the lowest lidar profiles, effectively helping solve the overlap issues that are often a challenge in lidar systems.

c. **Broad Applicability**: One of the key advantages of our approach is its only reliance on routine in situ measurements to derive aerosol vertical optical properties. This principle is applicable to other sites. As part of our future research efforts, we aim to collect and analyse data from additional sites to conduct broader and more comprehensive studies.

d. **Validation**: While our primary aim was not to apply this method directly to CAELI observations, it nevertheless offers a significant reference values for validating observations made by CAELI or similar instruments.

Moreover, while the literature suggests that traditional methods can provide reasonably reliable lidar ratio values, the effectiveness of these traditional approaches is inherently limited to sites equipped with Raman lidar measurement capabilities. This limitation is a significant constraint, as not all atmospheric research sites possess such advanced instrumentation. Our methodology offers a valuable alternative for deriving lidar ratio through conventional in situ measurements, significantly broadening the applicability of aerosol research to sites without advanced lidar technology.

2.   The presentation is rather unfocussed with an additional 42 figures in the supplement. The
authors also mention instruments like the microwave radiometer and the ceilometer that are
not really used later in the work. I suggest to identify key messages and trim the
presentation accordingly.

We have undertaken a thorough revision of our manuscript to address this issue. We have
now put some key information from the appendix to the main text, ensuring that the most
critical data and findings are readily accessible. Additional detailed profiles and
supplementary data (previously Figure S21-42) have been relocated to a publicly accessible
repository (https://doi.org/10.5281/zenodo.11174465), providing an opportunity for
interested readers to explore this information further.

Furthermore, we have clarified the role of the ceilometer in our study, specifically its utility
in identifying boundary layer heights, which is mentioned on line 193-195 in the revised
manuscript.

While the microwave radiometer data were not a primary focus of our investigation, the
meteorological data, particularly relative humidity measurements, play a crucial role in our
research. These data were incorporated for comparison purposes, underscoring that
ultimately, utilizing data from ECMWF is a superior choice, which also broadens the
applicability of our model. However, in the revised version of the manuscript, we have
chosen to retain only 3 profiles (Figure 6-8) to avoid overwhelming the readers with
excessive detail.

3.   The presented measurements seem to be quite specific focussing almost exclusively on
clean conditions. It would have been nice if there had been an effort to put the aerosol
conditions during RITA into a long-term perspective, e.g., using long-term sun-photometer
measurements.

In our original manuscript, we discussed two relatively polluted cases and two clean cases
to present a balanced view of aerosol conditions. In the revised version, we continue to
showcase results under both polluted (originating from the continent, 1$^{st}$ case and 2$^{nd}$ case)
and clean conditions (originating from the ocean, 3$^{rd}$ case). While the pollution levels in
these cases might seem mild when compared to the more heavily polluted regions globally,
they are representative of the pollution encountered in the Netherlands. This is
demonstrated in Figure R1. The boxplots below compare PM$_{2.5}$ concentrations from May
to October 2021 to the conditions encountered during the two RITA campaign periods
(Figure R1(b)) and the lidar case studies (Figure R1(c)). The cases discussed in the
manuscript are indicated by dashed grey lines.

[Figure]

Figure R1: Boxplot of PM$_{2.5}$ mass concentration (a) during the entire period from May to
October 2021 with outliers excluded; (b) Two intensive campaign periods (11$^{th}$ May - 24$^{th}$
May and 16$^{th}$ September - 12$^{th}$ October) with outliers excluded; (c) lidar vertical profile available periods with all data included. The three gray dashed lines represent the mass
concentrations for the three cases discussed in the revised manuscript.

Among all the lidar measurement periods, there was only one case where considerably
higher pollution values were recorded at the surface than in case 1 and 2, namely 18.6 µg
m$^{-3}$ as shown in Figure R1(c). This corresponds to the profiles showing below in Figure R2.
However, the layer above 1 km was very clean, and only a very small region with a valid
lidar ratio retrieval was available, therefore we did not focus on this case. However, within
the small region, predicted and retrieved lidar ratio agreed well. This case further
underscores the benefits of using in situ measurements to validate the reliability and
accuracy of lidar observations.

[Figure]

Figure R2: Profiles from 20:00 to 20:34 at UTC time on 2021-09-16.

Regarding the incorporation of long-term measurements to provide a broader perspective
on aerosol conditions during the RITA campaign, we acknowledge the value of such an
analysis. However, the availability of sun-photometer measurements during the RITA-
2021 campaign period was very limited, which constrained our ability to conduct a more
extended analysis. Despite this limitation, integrating long-term data into our research is
indeed part of our future plans.

4. The authors admit that coarse-mode aerosols have a large impact on the scattering
calculations. While this is somewhat minimised by the clean conditions considered in their
work, it is likely to be of huge importance during other conditions. In that context, it would
have been nice to get some long-term perspective on the occurrence of coarse-mode
aerosols. The authors should also mention that Mie theory is inadequate to infer optical
properties of dust particles.

In our study the mass fraction of coarse mode (PM$_{2.5-10}$) vs PM$_{10}$ aerosols was about 49%
on average, ranging from 14% to 81% during the two RITA campaign periods (11[th] May -
24[th] May and 16[th] September - 12[th] October in 2021). This is in line with previous longer-
term measurements of PM$_{2.5}$ represented 50 % of the PM$_{10}$ mass fraction at Cabauw from
September 2007 to October 2008 (Mensah et al., 2012). The coarse mode accounted for
about 13% of the total light scattering on average (ranging from 0.05% to 94.5%) in our
study. Under these conditions, our method can still provide reasonably accurate lidar ratio
predictions even in the absence of specific chemical composition data for coarse-mode
particles, as long as the size distribution of the coarse mode is known. Although in
individual cases, there are sometimes significant differences in predicted and retrieved lidar ratios in the original manuscript (as the dates are indicated in the Figure R3(a) below), we
found the coarse mode mass fraction was about 10% to 20% during these specific cases, so
the difference were not due to the coarse mode. In fact, the comparison has been improved
when we applied the overlap correction function to our retrieved data (Figure R3(b)) in the
revised manuscript. This suggests that in our study, the mismatches between the model and
the lidar measurement were seldom due to coarse mode but rather to the lack of overlap
correction.
This is a significant result and we aim to apply this method at other sites and conduct long-
term studies in the future to provide a more comprehensive understanding of aerosol
conditions, for which this applies. We expect at more inland sites this result might still be
valid, even if the coarse mode makes up a bigger mass fraction, because the chemical
composition is typically much less variable than the extreme cases (pure sea salt vs pure
mineral dust) we consider for the Cabauw site that is sometimes considerably influenced
by marine air masses.
We acknowledge the limitations of Mie theory in accurately predicting the optical
properties of non-spherical dust particles. This limitation is an important consideration in
future research, especially in regions where dust plays a significant role in atmospheric
aerosol composition.

[Figure]

Figure R3: Scatter plot of the lidar ratios from Raman lidar measurements (x axis) (a)
original data (b) new data with overlap correction applied, and from calculations (y axis) at
355 nm.
5.  The optical profiles inferred from the ground-based in-situ measurements all look like
scaled versions of the RH profile. This is not surprising as only RH might give some insight
on vertical variation and the authors assume aerosol conditions to be constant with height.
It doesn't seem fair that any discrepancies between measured and modelled optical
properties are then attributed to errors in the RH profile. The authors should rather find a
way to identify a maximum height up to which the vertical extension of ground-based in-
situ measurements can give meaningful results. High-resolution sounding profiles could be
a source for such an assessment and I wouldn't assume any connection between the ground
and above the first inversion - particularly later in the year.

In our methodology and results, relative humidity (RH) indeed plays a significant role in
the outcomes of the vertical profiles. However, as stated in our original manuscript,
discrepancies between the model and lidar measurements are not solely attributed to
variations in RH. Other factors, such as different aerosol layering (line 439-441) and
aerosol shape effects (line 426-429 and line 463-466), can also significantly contribute to
overestimation or underestimation of the backscatter and extinction profiles. Indeed, these
extensive aerosol properties are not directly predicable from ground-based measurements above the mixed layer. Nonetheless, an intensive property, such as the lidar ratio is still comparable between retrievals at around 1 km and the ground, if that height is within the boundary layer or residual layer.

The maximum height up to which the vertical extension of ground-based in-situ measurements (in particular the lidar ratio) can yield meaningful results was determined based on ceilometer data, as described in the original text (line 194-196). In the revised version of our manuscript, we have emphasized this point (line 193-195) to clarify the basis our methodology.

6. The authors overstate their findings. They state "a representative lidar ratio can be estimated based on ground-based in-situ measurements". However, the presented results give the impression that they are by no way superior to an analysis by an average lidar operator. The suitability furthermore hinges on the assumption of vertical homogeneity which - though not unreasonable - should still be supported by some form of measurement. They continue "This allows to extend extinction profiles to lower altitudes, where they cannot be retrieved, or for use with simple elastic backscatter lidar to derive extinction profiles." Again, lidar operators have been doing pretty well with assuming lidar ratios based on experience. The authors would need to be more specific to support their statement. And conclude "The proposed method could be further applied to predict aerosol optical depth and also might be beneficial for large-scale or global radiation simulations." It is quite customary to simply assume constant lidar extinction coefficients from the lowermost trustworthy measurement height to the surface. This approach generally shows good agreement to Sun-photometer observations of aerosol optical depth. This approach also assumes vertically homogeneous aerosol conditions but is much more straightforward than the authors' work.

In addition to the points highlighted in response 1, we wish to reiterate the significant contribution of our research to the atmospheric science community.

Our study effectively bridges the gap between in situ measurements and remote sensing lidar observations, addressing a notable gap in the literature where few studies have integrated these two independent methodologies for vertical aerosol profiling.

Our approach extends beyond the conventional column-integrated optical depth (Sun-photometer observations) closure to provide a broad exploration of vertical aerosol profiles. This allows for a deeper understanding of aerosol optical properties across different atmospheric layers.

Additionally, we believe that science should embrace multiple perspectives rather than be confined to or satisfied with a single methodology. By integrating different methodologies and independently verifying the results, we can improve reliability of our measurements.

In conclusion, we believe that the integration of in situ and remote sensing measurements is definitely worthwhile, offering a more accurate assessment of aerosol optical properties and their spatial distribution. This methodological innovation aligns with ACP's scope of enhancing our understanding of the atmosphere's composition and its broader impacts.

7. The choice of references regarding the lidar technique in general and lidar ratios in particular is quite unusual. I suggest to consult with the corresponding co-authors to find more suitable references.

We have consulted with our co-authors and have revised our reference list (line 45-59) to include more appropriate and widely recognized sources in the field.

References:

Mensah, A. A., Holzinger, R., Otjes, R., Trimborn, A., Mentel, T. F., Ten Brink, H.,
Henzing, B., and Kiendler-Scharr, A.: Aerosol chemical composition at Cabauw, the
Netherlands as observed in two intensive periods in May 2008 and March 2009, Atmos.
Chem. Phys., 12, 4723–4742, https://doi.org/10.5194/acp-12-4723-2012, 2012.

---

## Author Comment (AC2)

Response to referees for manuscript "Aerosol optical properties within the atmospheric boundary layer predicted from ground-based observations compared to Raman lidar retrievals during RITA-202" by Xinya Liu, Diego Alves Gouveia, Bas Henzing, Arnoud Apituley, Arjan Hensen, Danielle van Dinther, Rujin Huang, and Ulrike Dusek. (manuscript ID: EGUSPHERE-2023-2262)

We would like to thank the referee for the valuable comments on our paper, we believe that the manuscript has been improved significantly due to their suggestions. To facilitate the review process, we have copied the comments in black text and renumbered them for easy cross-referencing. Our responses are in standard blue text. We have responded to all the comments made by the referee and have revised the manuscript accordingly.

Referee #1 comments:

**1   General remarks:**

The manuscript discusses an interesting approach and brings together in situ observations of microphysical aerosol properties and chemical composition at ground (at Cabauw in The Netherlands) and optical modeling and comparison of the modeling results with lidar profiles of measured aerosol optical properties. There are many examples of such so-called closure experiments in the literature (since about 25 years), however, such exercises are still needed and thus the manuscript is a good addition to the literature in this field of optical closure studies.

In contrast to the other reviewer, I do not think that the paper should be rejected. It shows the present state of the art when combining ACTRIS observations from super sites… equipped with (a) aerosol monitoring tools and (b) remote sensing instruments. I also do not agree that this manuscript is a measurement report. Closure studies as presented here are more than just observations.

My main point of criticism is the following one: A lot of essential information is given in the rather extended supplement. That means one has to be very 'active' as reader and switch from main text to supplementary material and back and so one. A fluent reading is not possible. In the detail section, I will provide a few suggestions how this can be overcome, at least a bit.

Thank you for acknowledging the significance of our study and providing valuable suggestions for improving our manuscript.

In response to your comments, we have streamlined the main text by integrating critical information from the supplementary material directly into the main manuscript, wherever feasible. In addition, in order to minimizes the amount of supplementary material while still providing an access to readers who interested in this information, we have transferred the FigureS21-42 in the original manuscript to a public repository (https://doi.org/10.5281/zenodo.11174465).

Furthermore, we have carefully considered your detailed suggestions and have made corresponding adjustments to further improve and clarify our study.

We hope that these revisions address your concerns and look forward to your further guidance and feedback.

**2   Details:**

2.1   The Abstract needs to be adjusted after finalization of the revision.

We have carefully revised the abstract to better reflect and summarize the contents of our manuscript.

2.2 p2, l53: Cooney et al. and Melfi references are not appropriate here, in the context of aerosol extinction retrieval… Cooney and Melfi are pioneers in the field of Raman lidar developments because they introduced the temperature and water vapor Raman lidar technique.

We have revised our manuscript to correct this oversight (line 53) and ensure that our references accurately reflect the context of cited works. Thank you for bringing this to our attention.

2.3 p3, l81: be more specific already here, mention time periods.

We have adjusted accordingly and the time period is stated at the beginning of the methodology description on line 81.

2.4 p3, l84: I would prefer to include Figure S1 in the main text, and even Figure S2.

We have combined Figure S1 and S2 into a single figure, which is now included in the main text as Figure 1.

2.5 p5, l144: Avoid confusion (with lidar backscatter at 180°) already in the beginning, mention the angle range directly after … backscatter coefficient (7° to 170°).

We now mentioned the angle range (7° to 170°) directly after introducing the backscatter coefficient, to avoid any potential confusion with lidar backscatter at 180° on line 146. Thank you for highlighting this point.

2.6 p6, l174: When having a near range telescope you should be able to show extinction and lidar ratio values even down to 500 m height (after overlap corrections). And for heights above about 1000 m, you should be able to use the far range observations (after overlap correction) and then we would have much better, less noisy lidar ratios between 1000 and 2500 m height. So, why are these data not included? ….. should be stated…. I would recommend to use your own Raman analysis method instead of using the automated SSA software, and in this way, to optimize the lidar products in these optical closure studies.

We thank the referee for these observations and suggestions. Encouraged by this comment, we investigated the application of an overlap correction to reduce the minimum valid altitude for the extinction and lidar ratios. For that, the overlap function has been determined using the method proposed by Wandinger and Ansman (2002) from measurements with relatively low aerosol loads (ext. coef < 100 Mm-1, clean/no residual layer), and the results were used to reprocess the data. The average overlap correction found agrees with our expectations based on the telecover tests, being < 3% for ranges above ~700m. We found that the overlap correction improved the retrievals, allowing us to reduce the minimum valid altitude for all cases down to 810 m, and also demonstrating a better match with the model calculations. Examples are provided below (Figure R1-R3), showing the comparison of profiles before and after overlap correction with model results. For heights below this altitude, the overlap vertical gradient increases rapidly and the uncertainty of the a-priori lidar ratio becomes more important, even for clean days. To avoid these issues, we keep the minimum valid altitude as 810 m and we have incorporated the reprocessed data to the revised manuscript.

[Figure]

Figure R1: Comparison of profiles before (blue line) and after (orange triangle) overlap correction with model results (blue dot) from 21:12 to 21:30 at UTC time on 2021-05-12.

[Figure]

Figure R2: Similar to Figure R1 but showing profiles from 20:25 to 21:25 at UTC time on 2021-07-08.

[Figure]

Figure R3: Similar to Figure R1 but showing profiles 21:12 to 22:00 at UTC time on 2021-07-19.

Regarding the far field range telescope (FFR) not being used for the retrievals: The Cabauw station has been prioritizing lidar processing using the Single Calculus Chain (SCC) in a networked effort for centralized, harmonized and quality assured data processing in the framework of ACTRIS/EARLINET. Unfortunately, the SCC cannot yet combine the near and far field telescopes in its retrievals. That is a more practical reason why only the near field range telescope (NFR) was used in the retrievals. Although it is true that the combination of the near and far field telescopes would yield optical products with better signal-to-noise ratios, the retrieved extinction profiles below ~1500 m wouldn't be greatly different at 355 nm and the SNR improvement on the 532 nm extinction would still be insufficient to compensate for the low pulse energy we had for the visible wavelength during the campaign. This reduces the added value of including the FFR signal for this work. For those reasons, we will remain using the Single Calculus Chain for the backscatter and extinction retrievals in the revised paper.

The paragraph describing the lidar processing (section 2.3.1) was changed accordingly in line 178-184 of the revised manuscript

2.7 Another question: What about the Raman lidar observations of the water vapor mixing ratio. In combination with ECMWF temperatures (usually very accurate) one could present them in the panels with ECMWF T and RH profiles. Even during daylight conditions, I could imagine that signals are good enough to show water vapor data up to 1000 m height.

The ECMWF RH profiles are rather uncertain (as usual for modelled water vapor profiles), so one needs more information about the 'real world' RH conditions. I would appreciate, if one shows radiosonde profiles in the respective panels in Figures 5,6,7,8, and if possible the Raman lidar RH profiles. The humidity has such a large and critical impact on the modelled optical properties, one needs to show better RH values, even if ECMWF RH values are considered in all the modelling, the reader should know about the quality (uncertainty) of these ECMWF RH profiles.

Thanks for your suggestions. We recognize the importance of accurately representing real-world RH conditions and the critical impact of humidity on modelled optical properties, thus, we have included radiosonde profiles and Raman lidar water vapor mixing ratio derived RH profiles in Figure 6-8 in the revised manuscripts. This addition is aimed at providing readers with a clearer understanding of the quality and uncertainty associated with ECMWF RH profiles.

2.8 Figure 1 is certainly confusing for non-lidar scientists, especially regarding all the vertical white lines up to 5 km height. Is that just noise or is that strong backscatter from clouds…? Furthermore, what do you mean: an overview is given in Figure 1…., when nothing is explained? What is then the message to the reader? The Raman lidar observations need to be better indicated by thicker lines and brighter color, maybe yellow or orange.

We apologize for any confusion caused by the initial demonstration of Figure 1. In response to your feedback, we have revised the description of Figure 1 (line 195-198) to clarify its content and purpose. Additionally, we have improved the visualization of Raman lidar observations within the figure by employing thicker lines and brighter colors.

2.9 p9, l232: I would include Table S3 in the main manuscript.

Response: Thank you for your advice. We acknowledge the significance of the data in Table S3. Accordingly, we have incorporated Table S3 into the main text as Table 1 in the revised manuscript.

2.10 p9, l243: One should better highlight and explain, how the vertical profile is obtained….
Maybe, one should have a subsection (on vertical aerosol profile) , and show a sketch…,
showing T and RH profiles, a well-mixed PBL, and maybe even T and RH profiles for a
well-mixed layer, i.e. pot temp = const, RH increasing according to mix ratio = const. In
addition, the optical properties as modelled at the surface (indicated by a big symbol)
should be shown and finally the aerosol extinction profile, that is in agreement with the
RH height profile structures.

Such a sketch would support the reader to understand the closure results…. in Figs. 5-8.

Thank you for your suggestion. In response, we have improved the depiction of the
calculation flowchart and made a sketch for the vertical profile calculations, now
presented as Figure 3 in the revised manuscript. We added a much more detailed
explanation of how the vertical profile is obtained, with all the relevant equations along
with the reasoning behind it in section 2.4 of the manuscript.

**3   Results and discussion:**

3.1   I would prefer to start with Figure S1 and S2 in the main text! Four case studies are then
discussed. To provide all necessary details (to the field site, trajectories etc…) in the main
text, one probably has to reduce the number of case studies. In the case of Figure 5, I
would prefer to see in addition Fig. S9 (showing the full advantage of a lidar, clearly
indicating many different aerosol layers, rather than any well-mixed layer), Fig. S10,
providing information about the chemical composition, and Fig. S11, showing the origin
of the pollution. However, we need at least different trajectories at 250 m (representative
for surface aerosol conditions), 900 m, and also one for the 1200-2500m aerosol layer.

In this way (Fig. 5, S9, S10, S11), we would have a complete story and could much better
discuss the results of the closure study, and why there is disagreement, especially for
heights above 1200m.

I also believe that a full set of observed information (including a much better description
of the humidity conditions and air mass transport at different heights) will allow a critical
and much deeper debate on the applicability of the closure approach presented here and
the especially concerning the limits of the approach.

And as mentioned, I would include a nighttime radiosonde RH profile (19 May, 23:30
UTC, Figure S6 shows it), and if Raman lidar mixing ratio data are available even Raman
lidar based RH profiles.

Fig. 5: I do not see (a), (b), (c), (d), where did you put/place these letters? If there are only
355 nm extinction and lidar ratio profiles, then one should not show 532nm in the boxes
(with line and symbol explanations), and these white boxes should not hide values. This
holds for all other figures and panels as well.

Thank you for your comprehensive feedback, which has guided us to make thorough
adjustments to present our story more coherently.

Firstly, we have integrated Figures S1 and S2 into the main text within the methods
section 2.1 to provide readers with a clearer understanding of the experimental setup from
the outset.

Following your suggestion, we have combined the content of Figures 5, S9, S10, and S11
into a single figure (Figure 6 in the revised manuscript). This allows readers to grasp the
entirety of a case study without the need to navigate between the main text and
supplementary material.

As per your recommendations, we have incorporated the lidar derived RH profile, radiosonde RH profile, ECMWF profiles into a single plot as shown in Figure 6(d) in the revised manuscript. We also added backward trajectories at 3 different heights (Figure 6(f)) for providing a more comprehensive information.

Finally, we have improved the visualization of the original image, adding necessary numbers and preventing valid information from being hidden. Subsequent pictures have also been adjusted accordingly.

3.2 The same for Figure 6, we need in addition, Fig S12, S13, and 14 (with three trajectories) in the main text. And on 9 Sep, it was probably dark over Cabauw at 21 UTC…. so please show Raman lidar RH profiles plus radiosonde RH profiles (9 Sep 23:30 UTC).Now, we can discuss this closure study in very large detail, including the uncertainty in the model results caused by the ECMWF RH profile.

Similar to the previous response 3.1, we have made the corresponding changes in the revised manuscript.

3.3 I would skip the Fig. 7 closure study. There is already the 19 May case, and the lidar ratio shows marine conditions. Figure 8 is nice, could be presented with the figures S18-S20 here in the main manuscript, and again more trajectories for more heights (250 m, 800m, 1600 m) should be shown. Furthermore, Raman lidar and radiosonde water vapor profiles, if available. Alternatively, one could try to combine Figs. 7 and 8 in ONE figure and show only the optical properties, and briefly discuss the results of these closure study.

In the revised manuscript, we have removed original Figure 7 and provided a detailed discussion on only one clean case (Figure 8). The corresponding modifications are in the revised version.

3.4 Figure 9 shows just ONE 532 nm lidar ratio. I would remove this 532 nm value, so that only measured 355 nm lidar ratios are considered in Fig 9a and 9b.

In response to your suggestion regarding Figure 9, we have removed the retrieved lidar ratio at 532 nm to avoid any potential confusion.

3.5 The supplementary material is too much, no reader (except the reviewers) will study all details so one should reduce the amount of figures and plots to an absolute minimum.

Thanks for the suggestions. We have taken steps to streamline the content, reducing the number of figures and plots to an essential minimum. However, considering the potential value of these materials to interested readers, we have relocated the additional content to a publicly accessible repository (https://doi.org/10.5281/zenodo.11174465).

Ref.: Wandinger, U. and Ansmann, A.: Experimental determination of the lidar overlap profile with Raman lidar, Appl. Opt., 41, 511, https://doi.org/10.1364/ao.41.000511, 2002.

---

## Referee Report (RR1)

Review of "Aerosol optical properties within the atmospheric boundary layer predicted from ground-based observations compared to Raman lidar retrievals during RITA-2021" by Xinya Liu et al, submitted to Atmospheric Chemistry and Physics

I read the reviewer comments and the replies of the authors.

To my opinion the points the reviewers had raised have been addressed adequately well in the revised version. The revised version has undergone (in part) substantial text changes which largely clarify the challenges that come with using ground-based point observations (dry atmospheric conditions in the particle sampling chamber) for the task described in this paper. The use of Raman lidar profiles of backscatter and extinction measured at ambient atmospheric conditions can be considered as the benchmark. The revised version presents a good, critical evaluation of the results and it includes comments on where the methodology presented in this paper stops being useful. I think readers who possess the necessary knowledge in lidar data analysis/ lidar instrument development will gain new insight into the challenging task of using in-situ ground-based observations in two areas of atmospheric observations with lidar and in-situ instruments:

1) How to extend lidar profiles from the first height bin where lidar profiles become useful (full overlap between laser beam and receiver-field-of-view is achieved) to the ground (no lidar data or data affected by an incomplete overlap).
2) How to make better use of data collected with elastic-backscatter lidar, how to use relative humidity profiles and modelling, and how to combine all this information together with in-situ observations in a smart way.

I furthermore consider the use of the SCC as an essential part of the work presented in this manuscript. The use of the SCC understandably means that certain compromises need to be accepted in terms of what type of data are available and how this comparison study can be carried out. I think a good compromise has been found in terms of what needs to be done if "the last bit of information" is squeezed out from the data and the methodology, and what can realistically be achieved (by using SCC in a semi-automated mode). SCC is a key tool of working with lidar in ACTRIS. Yet, even though the authors' methodology shows reasonable results I encourage them to "squeeze harder" to further improve their methodology.

I also want to emphasize another reviewer's comment that it is necessary to have more publications on this topic of combining in-situ ground-based data with lidar observations. There are few observation sites that have the necessary hardware and data-analysis experience to carry out such observations in the first place.

My opinion is that the "lidar" profiles (from the modelling) seem to follow too much the relative-humidity profiles, i.e. the modelled lidar profiles may be biased. This point has been raised by one of the reviewers in the first round of reviews. However, I do not see a reason for unwrapping everything at this stage. The replies of the authors to the first round of reviews addresses this point.

For all the above reasons I therefore ask for minor changes/correction:

- I suggest the abstract contains a bit more emphasis on the need of "... a well-mixed PBL including accurate corrections for hygroscopic particle growth" and to emphasize the fact that "... the relative humidity profile may have substantial influence on the shape of the profiles."
- I am asking the authors to become crystal clear on the fact that particle loss effects during sampling of coarse mode particles by the in-situ instruments could become a serious issue if more coarse mode particles are present. Make a respective statement in the conclusion section, please.
- Line 44, the word "of": is it missing in "... spatial distribution aerosol ..."?
- Line 72: please consider adding a bit more text to "... the extinction coefficient in the lower atmosphere ...", e.g.by adding "... in the region of incomplete overlap between laser beam and receiver field-of-view of the lidar detector system ..."
- You need to add a short outline of your paper at the end of the introduction section (section 2 ..., section 3, ... etc). I think that is (still) a standard part of a scientific publication work.
- Lines 140/141 "... the MPSS electrical mobility diameters were assumed to correspond 140 to volume-equivalent diameter.": can you please insert a reference that corroborates this comment?
- line 186: you need to add the pulse energy to the information on the laser.
- Figure 3: there is a typo in "Tempeture". Please also add (in the figure caption) the meaning of the abbreviations. Please explain (in the legend) the meaning of SIA, EC, SS, MD. Please explain in (3a) CC, RH, GF, PSD, no matter whether it is mentioned in the main body of the text or not. Essential information needs to be given in figure captions, too.
- Line 215: MD? I am curious where mineral dust should come from? Or do you refer to road dust, agricultural dust etc.? I repeat that I do not intend to re-open the review process, but rather consider using backward trajectories in a more extensive way in your future work. 72-hours backward does not tell you a lot. Go for 5-days backward (at minimum). 10 days may be even better. The vertical movement of the trajectories is important, too. It tells you a lot about the possible source of particles.
- Line 280: I think it is nephelometer, not Nephelometer.
- Line 248: number is missing in "approximately ... Mm^-1".
- Please check formatting issues like the following: often numbers are followed by the unit "m" without space, sometimes with space. Make it the same.
- Line 413: add "sr" to "... 1.1 sr at 532 nm."
- Line 450: check the reference author name "Moritzet et al. "? Is it "Moritzet" or is it "Moritz et al."? In fact, I do not find any reference with that name in the reference list. In other words: check your reference list for completeness.

---

## Author Response (AR2)

Response to referees for manuscript "Aerosol optical properties within the atmospheric boundary layer predicted from ground-based observations compared to Raman lidar retrievals during RITA-202" by Xinya Liu, Diego Alves Gouveia, Bas Henzing, Arnoud Apituley, Arjan Hensen, Danielle van Dinther, Rujin Huang, and Ulrike Dusek. (manuscript ID: EGUSPHERE-2023-2262)

We would like to express our sincere gratitude to the reviewers for their careful reading and helpful comments on our manuscript. We have revised the manuscript and responded to each comment below. We appreciate the opportunity to refine our research and look forward to the next steps in the publication process.

Reviewer #1 comments

 I found several minor points that should be improved:

1. The sketch in Figure 3 Tempeture is written

    Corrected.

2. In the main text body, page 11, line 282… Figure 2(b) is written. but it is Figure 3(b). Please check numbering of figures and table in the main text carefully.

    Corrected.

3. Figure 6(d): It took me a while to identify the different curves. The legend should be more clear. The blue line (without symbols) is the Raman lidar water vapor profile, in the legend is light blue, like the color of the uncertainty margin! The light blue uncertainty margin obviously belongs to the RH ECMWF profile. This is not obvious from the legend. The dark blue RH ECMWF profile has symbols (circles), but this is not shown in the legend. The RH tower observation are shown as symbols (stars) in the legend, but these points are hard to find in the figure. So please try to improve all this, one can even use open symbols (circles and squares) , dashed and dotted lines. There are many options to improve the figure. The same holds for Figure 7(d) and 8(d).

    Thank you for your suggestions. We have modified Figure 6(d), Figure 7(d), and Figure 8(d) to improve visualization.

Reviewer #2 comments:

1. I suggest the abstract contains a bit more emphasis on the need of "… a well-mixed PBL including accurate corrections for hygroscopic particle growth" and to emphasize the fact that "… the relative humidity profile may have substantial influence on the shape of the profiles."

    Thank you for your advice. We have added relevant text in the abstract (lines 23-24) to emphasize these two important points you mentioned.

2. I am asking the authors to become crystal clear on the fact that particle loss effects during sampling of coarse mode particles by the in-situ instruments could become a serious issue if more coarse mode particles are present. Make a respective statement in the conclusion section, please.

    In the revised manuscript, we have added text in lines 469-471 and lines 496-498 to highlight the importance of coarse mode particles.

3. Line 44, the word "of": is it missing in "… spatial distribution aerosol …"?

    Corrected.

4. Line 72: please consider adding a bit more text to "… the extinction coefficient in the lower atmosphere …", e.g.by adding "… in the region of incomplete overlap between laser beam and receiver field-of-view of the lidar detector system …"

Done.

5. You need to add a short outline of your paper at the end of the introduction section (section 2 …, section 3, … etc). I think that is (still) a standard part of a scientific publication work.

We provide a brief introduction to each section at the end of the introduction (lines 78-83 in the revised manuscript).

6. Lines 140/141 "… the MPSS electrical mobility diameters were assumed to correspond 140 to volume-equivalent diameter.": can you please insert a reference that corroborates this comment?

We have added a reference at the place.

7. line 186: you need to add the pulse energy to the information on the laser.

Done.

8. Figure 3: there is a typo in "Tempeture". Please also add (in the figure caption) the meaning of the abbreviations. Please explain (in the legend) the meaning of SIA, EC, SS, MD. Please explain in (3a) CC, RH, GF, PSD, no matter whether it is mentioned in the main body of the text or not. Essential information needs to be given in figure captions, too.

We corrected the typo and explained the meanings of the abbreviations in the Figure caption.

9. Line 215: MD? I am curious where mineral dust should come from? Or do you refer to road dust, agricultural dust etc.? I repeat that I do not intend to re-open the review process, but rather consider using backward trajectories in a more extensive way in your future work. 72-hours backward does not tell you a lot. Go for 5-days backward (at minimum). 10 days may be even better. The vertical movement of the trajectories is important, too. It tells you a lot about the possible source of particles.

The contribution of mineral dust (MD) to particulate matter concentrations in the Netherlands is relatively minor. Our research references data from the 2019 Trolix campaign (14$^{th}$ Sep to 6$^{th}$ Oct) conducted at the same sampling sites Cabauw, where MD accounted for approximately 30% of the mass concentration of coarse mode particles, with an average concentration of about 0.45 µg/m$^3$. Previous studies have found the primary sources of MD in the Netherlands are road dust (such as Tyre (mostly OC and Zn) and brake (Fe, Cu, Sn, Sb, Ba) wear processes) and agricultural activities (Amato et al., 2013; Hendriks et al., 2013).

We greatly appreciate the reviewer's suggestion to use longer backward trajectories to obtain more potential information on aerosol sources. To ensure the validity of our conclusions in this study, we have examined 10-day backward trajectories of the three cases in the manuscript and found that they are mainly influenced by similar sources as the currently used 72-hour backward trajectories. The Figure R1 below provides an example, illustrating a comparison between the 3-day backward trajectory and the 10-day backward trajectory for the first case on May 19, 2021, at 20:00. Both trajectories indicate that the predominant air mass transport was from the ocean to the land, passing through Ireland and the United Kingdom, ultimately reaching the monitoring site.

[Figure]

Figure R1: (a) 3-day backward trajectories (b) 10-day backward trajectories ending at 20:00 on 19-05-2021.

We will also take this into consideration in our future research work.

10. Line 280: I think it is nephelometer, not Nephelometer.

   Corrected.

11. Line 248: number is missing in "approximately … Mm^-1".

   Corrected.

12. Please check formatting issues like the following: often numbers are followed by the unit "m" without space, sometimes with space. Make it the same.

   Done.

13. Line 413: add "sr" to "… 1.1 sr at 532 nm."

   Done.

14. Line 450: check the reference author name "Moritzet et al. "? Is it "Moritzet" or is it "Moritz et al."? In fact, I do not find any reference with that name in the reference list. In other words: check your reference list for completeness.

   Corrected.

References:

Amato, F., Schaap, M., Denier van der Gon, H. A. C., Pandolfi, M., Alastuey, A., Keuken, M., and Querol, X.: Short-term variability of mineral dust, metals and carbon emission from road dust resuspension, Atmos. Environ., 74, 134–140, https://doi.org/10.1016/j.atmosenv.2013.03.037, 2013.

Hendriks, C., Kranenburg, R., Kuenen, J., van Gijlswijk, R., Wichink Kruit, R., Segers, A., Denier van der Gon, H., and Schaap, M.: The origin of ambient particulate matter concentrations in the Netherlands, Atmos. Environ., 69, 289–303, https://doi.org/10.1016/j.atmosenv.2012.12.017, 2013.